# COMRECGC: Global Graph Counterfactual Explainer through Common Recourse

**Grégoire Fournier** [1] **Sourav Medya** [2]

## Abstract

Graph neural networks (GNNs) have been widely used in various domains such as social networks, molecular biology, or recommendation systems. Concurrently, different explanations methods of GNNs have arisen to complement its black-box nature. Explanations of the GNNs' predictions can be categorized into two types—factual and counterfactual. Given a GNN trained on binary classification into "accept" and "reject" classes, a global counterfactual explanation consists in generating a small set of "accept" graphs relevant to all of the input "reject" graphs. The transformation of a "reject" graph into an "accept" graph is called a recourse. A common recourse explanation is a small set of recourse, from which every "reject" graph can be turned into an "accept" graph. Although local counterfactual explanations have been studied extensively, the problem of finding common recourse for global counterfactual explanation remains unexplored, particularly for GNNs. In this paper, we formalize the common recourse explanation problem, and design an effective algorithm, COMRECGC, to solve it. We benchmark our algorithm against strong baselines on four different real-world graphs datasets and demonstrate the superior performance of COMRECGC against the competitors. We also compare the common recourse explanations to the graph counterfactual explanation, showing that common recourse explanations are either comparable or superior, making them worth considering for applications such as drug discovery or computational biology.

## 1. Introduction

Graph Neural Networks (GNNs) have been widely used for graph classification across various domains such as electrical design (Mirhoseini et al., 2020), physical simulation (Bhattoo et al., 2022), or drug discovery (Jiang et al., 2020; Cao et al., 2016; Hamilton et al., 2017). Despite their popularity and their good performance, the predictions of GNNs are not yet fully understood, and explaining GNN's behavior has become a central focus of recent research efforts (Armgaan et al., 2024; Kosan et al., 2024; Kakkad et al., 2023; Yuan et al., 2023). Our work focuses on the concept of Counterfactual Explanation (CE) in the context of binary classification (e.g., 'reject' vs. 'accept' classes). This involves designing the *minimal changes* required in each 'reject', or undesirable graph, to be able to change to an 'accept' graph, or counterfactual graph, that is often similar in structure. A global CE, in particular, aims to identify a small set of 'accept' graphs relevant to all 'reject' graphs, thereby providing insights into the critical decision regions recognized by the base GNN model. This global explanation approach highlights the model-level behavior as opposed to an instance level or local one (Verma et al., 2024).

Another important aspect of CE explanation is the implicit provision of recourse—graph transformations that convert a 'reject' graph into an 'accept' graph (Verma et al., 2022; Karimi et al.). This is particularly valuable in applications where it is possible to take action to reverse a decision, such as in loan applications or in modifying a compound to change physical or chemical properties of molecules (e.g., mutagenicity (Riesen & Bunke, 2008)).

The instance-specific or local counterfactual explanations are insightful and capable of providing recourse for each instance (Abrate & Bonchi, 2021; Bajaj et al., 2021; Lucic et al., 2022; Tanyel et al., 2023; Chhablani et al., 2024). However, the explanations generated are generally large in number and not straightforward to interpret at the model level. Global counterfactual explainers have been proposed to address these issues (Huang et al., 2023; Ley et al., 2023). *We present additional related work in Appendix A.* In (Huang et al., 2023), the authors build global counterfactuals for GNNs. However, a major limitation of their approach is its generalizability when considering recourse. A single

[1]Department of Mathematics, Statistics and Computer Science, University of Illinois Chicago, Chicago, USA [2]Department of Computer Science, University of Illinois Chicago, Chicago, USA. Correspondence to: Grégoire Fournier <gfourn2@uic.edu>.

*Proceedings of the $42^{nd}$ International Conference on Machine Learning*, Vancouver, Canada. PMLR 267, 2025. Copyright 2025 by the author(s).

counterfactual can lead to significantly different recourse depending on the specific graph it explains, making it unsuitable as a local explanation (Verma et al., 2024). In (Ley et al., 2023), translation directions are learned by a counterfactual explanation algorithm and used with variable magnitudes to generate a global explanation. While this approach improves generalization, it decouples the process of identifying translation directions from fitting to a specific data distribution, which may limit robustness in some settings. Moreover, this method has not yet been applied to graph data.

To answer these shortcomings, our work considers the problem of *finding common recourse (*FCR*)* graph explanation, which seeks to generate a small set of recourse that is capable of converting every 'reject' graph into an 'accept'. This essentially means that one or more graphs use the same recourse to achieve a counterfactual. To the best of our knowledge, this problem has not been studied before from a graph machine learning perspective. FCR offers both high-level interpretability and instance level insights. Moreover, in some cases, more input graphs can be explained with fewer common recourse compared to global counterfactuals. Our main contributions are the following:

- **Novel problem setting.** We formalize the FCR problem. We prove that the FCR problem is NP-hard. We provide a generalized version of the FCR problem, called FC (Finding Counterfactual), and derive an approximation algorithm to a constrained version of FC.

- **COMRECGC**. We design the COMRECGC algorithm to extract high quality common recourse as an explanation, which provides a solution to both problems.

- **Experiments.** We experiment on real-world datasets and benchmark our algorithm against popular counterfactual explainers on the Common recourse task. COMRECGC outperforms the next best solution on the FCR problem by more than $20\%, 30\%, 20\%$, and $10\%$ in total coverage on the NCI1, MUTAGENICITY, AIDS, and PROTEINS datasets respectively. We show that COMRECGC common recourse explanation offers a comparable and sometimes higher coverage than the best baseline global counterfactual explanation.

## 2. Background and Problem Formulation

**Counterfactual Explanation.** A graph is defined as $G = (V, E)$, where $V$ is the set of vertices and $E$ the set of edges. Consider a binary graph classification model—such as a GNN—where the prediction function is be denoted by $\Phi$. A graph is predicted in the 'reject' class if $\Phi(G) = 0$ and in the 'accept' class if $\Phi(G) = 1$. For any reject graph $G$, we call a graph $H$ *counterfactual* if $\Phi(H) = 1$ and

$d(G, H) \leq \theta$, where $\theta$ is a given normalized distance value such that $0 < \theta < 1$ and $d$ is a distance function. In other words, we say $H$ *covers* $G$. In the rest of the paper, we consider our input to be solely constituted of reject or 'undesirable' graphs, denoted by $\mathbb{G}$, and we denote by $\mathcal{C}(G)$ the set of counterfactual graphs that cover $G$.

**Common Recourse.** Given a reject graph $G$ covered by $H$, we call *recourse* a transformation (a function, $r$) which converts $G$ into $H$. Note that the cost of the recourse is the distance between $G$ and $H$. Upon defining a common recourse, an obvious question is how to define it on different graphs. For instance, if the recourse is 'adding' a vertex connected to some specific vertices that do not exist in different graphs, it will lead to some sort of ambiguity. However, a GNN always gives a latent representation (embedding). Thus, we use graph embedding $z$ from the space of graphs into $\mathbb{R}^l$ for some constant $l$ to define common recourse. Note that these embeddings can also be used to measure the distance between two graphs, such as the graph edit distance (Ranjan et al., 2023).

Suppose $H$ covers $G$, and let $r$ be the recourse that turns $G$ into $H$ (i.e., $r(G) = H$), then we define the embedding of $r$ in the space of vectors of $\mathbb{R}^l$ to be $\vec{r}$, the vector from $z(G)$ to $z(H)$, i.e., $\overrightarrow{z(G)z(H)}$. Given two recourse $r_1, r_2$, we say that they form a common recourse if there exists $\vec{v}$, a vector in $\mathbb{R}^l$, such that $||\vec{v} - \vec{r_1}||_2 \leq \Delta$ and $||\vec{v} - \vec{r_2}||_2 \leq \Delta$, for a fixed $0 < \Delta < 1$. The idea is that there exists a center recourse which is close to both recourse and can be used as a summary.

*Please refer to the Appendix for the proofs of the Claims and Theorems of this section.*

### 2.1. The Problem of Common Recourse

Our goal is to have a small representative set of common recourse, $\mathbb{F}$, and the cost of these should be small. Before formalizing our problem, we define the following:

- COVERAGE$(\mathbb{F}) := \big|\{G \in \mathbb{G} | \exists H \in \mathcal{C}(G), \exists r \in \mathbb{F}$ such that $||\vec{r} - \overrightarrow{z(G)z(H)}||_2 \leq \Delta\}\big|/|\mathbb{G}|$; the fraction of the input graphs for which at least one counterfactual is obtained through one of the recourse in $\mathbb{F}$. Note that the center recourse are being chosen from $\mathbb{F}$.

- cost$(\mathbb{F}) := \text{AGG}_{G \in \mathbb{G}}\{min\{ ||\vec{r}||_2, r \in \mathbb{F}$ and $\exists H \in \mathcal{C}(G)$ such that $||\vec{r} - \overrightarrow{z(G)z(H)}||_2 \leq \Delta\}\}$; the total distance from the covered input graphs to their closest attained counterfactual through $\mathbb{F}$. The AGG function used in the experiments is the summation.

- size$(\mathbb{F}) := |\mathbb{F}|$.

***Problem*** 1 (Finding Common Recourse (FCR)). Given input graphs $\mathbb{G}$ and a budget $R$, the goal is to find a set of

recourse of size $R$ that maximizes the coverage:

$$max_{\mathbb{F}} \text{ coverage}(\mathbb{F}) \text{ such that size}(\mathbb{F}) \leq R.$$

Recourse are derived from the nearby counterfactual graphs. For a set of counterfactuals $\mathbb{H}$, we define its associated recourse set as $\mathbb{F}_{\mathbb{H}} := \{\overrightarrow{z(G)z(H)}|\ G \in \mathbb{G}, H \in \mathbb{H} \cap \mathcal{C}(G)\}$. Subsequently, another way to formalize the problem would be to look at the common recourse that can be extracted from a set of counterfactuals.

***Problem*** 2 (Finding Counterfactual (FC)). Given input graph $\mathbb{G}$ and two budgets $R$ and $T$, the goal is to find a set of counterfactuals $\mathbb{H}$ such that its associated common recourse set has size $R$:

$$max_{\mathbb{H}} \text{ coverage}(\mathbb{F}_{\mathbb{H}}^{*}) \text{ s.t. size}(\mathbb{F}_{\mathbb{H}}^{*}) \leq R, size(\mathbb{H}) \leq T,$$

where $\mathbb{F}_{\mathbb{H}}^{*}$ denotes the common recourse set that achieves the maximum coverage from selecting $R$ recourse of $F_{\mathbb{H}}$.

Note that the FCR problem is a specific case of the FC problem by setting $T$ to $|\mathbb{G}|$, the number of input graphs to cover. The FC problem corresponds to a max multi-budget multi-cover problem, where we have separate budgets for counterfactuals and for recourse. To count an input graph as covered, it must be covered by both of these budgets. *We give an example for the* FC *problem in Appendix B.*

Although the cost is not a constraint nor an objective in both the FCR and FC problems, it remains, alongside coverage, a valuable metric for assessing the relevance of counterfactual explanations (Rawal & Lakkaraju, 2020), and by extension, the recourse explanation derived from them.

## 2.2. Analysis: The FCR problem

The FCR problem (Problem 1) consists in finding the best common recourse set of size $R$ in terms of coverage from a list of common recourse.

***Theorem*** 1. The FCR problem is NP-hard. (Appendix C.1)

Let us define $f$ as the function that associates to a set of common recourse its coverage. We claim that $f$ is monotone and submodular (Appendix C.2), hence the common greedy algorithm yields a $(1 - 1/e)$ approximation to selecting the $R$ best recourse, which is the best poly-time approximation unless P=NP (Feige, 1998).

## 2.3. Analysis of the FC problem

The FC problem (see Problem 2) is an extension of the FCR problem where we are no longer given the full common recourse set, and we are required to pick at most $T$ counterfactuals to form $R$ recourse for best coverage. In Appendix C.3, we present a budget version of the FC problem.

Let $g$ be the function that associates to a set of counterfactuals its best coverage through building common recourse. It

is not hard to see that $g$ is monotone and does not possess any local submodularity ratio (Santiago & Yoshida, 2020).

***Theorem*** 2. $g$ is not pseudo-modular. (Appendix C.4)

### 2.3.1. APPROXIMATION FOR THE FC PROBLEM

Since the function $g$ is monotone but not-pseudo modular (Theorem 2), finding an approximation algorithm with some guarantees is non trivial. To make the problem tractable for approximation, we add the following constraint:

**Constraint 1:** (**C1**) To be considered as a valid solution, a set of $T$ counterfactuals, $G_1 \ldots . G_T$ must verify:

$$\forall p < T,\ g(\bigcup_{i \leq p+1} G_i) - g(\bigcup_{i \leq p} G_i) > 0. \tag{1}$$

Intuitively, this means that there exists a "series" (in the sense of the union of sets) of counterfactuals that increases the objective function at each step. Such a constraint does not seem too abstract, as most approximation algorithms look to add one element at a time. We obtain the following:

***Theorem*** 3. There is a $1 - e^{-1/R}$ approximation algorithm in expectation for the FC problem with **C1**. (Appendix C.5)

**Discussions.** The above theoretical results have important implications: (**1**) **FCR problem:** The $(1 - 1/e)$ approximation guarantee of the greedy algorithm consists in selecting the best common recourse at each step once the counterfactuals have been found. It is featured in our method for solving the FCR problem. (**2**) **FC problem:** The $1 - e^{-1/R}$ approximation for the FC problem might be less useful in practice, as typically we want to allow numerous ($R \approx 100$) recourse to explain a dataset. Although the corresponding randomized greedy algorithm provides a guarantee, instead of using this algorithm, we will assess the importance of each counterfactual with its number of visits through a "common recourse random walk".

## 3. Our Method: COMRECGC

To find good counterfactual for common recourse, COMRECGC operates in different stages.

- **A graph embedding algorithm**: First, COMRECGC maps each graph from the set of input graphs into $\mathbb{R}^l$, an embedding space. From the representation of an input graph and one of its counterfactual, we derive the recourse embedding.

- COMRECGC finds counterfactuals for common recourse through a **multi-head vertex reinforced random walk** in the graph edit space. This is a variation of a vertex reinforced random walk (Pemantle, 2004).

- A **clustering algorithm** for common recourse: we form clusters over the embeddings of the generated recourse. Each cluster of a certain radius ($\Delta$) corresponds to a *common recourse*. Finally, we aggregate the common recourse greedily to obtain the maximum coverage.

### 3.1. Graph embedding algorithm

One of the essential notions for defining the counterfactual is the distance function, as mentioned earlier. Our method, COMRECGC, begins with the efficient computation of distance (or similarity) between two graphs. To assess distance between graphs, we use the graph edit distance (GED) (Sanfeliu & Fu, 1983) that accounts for the minimal number of transformations—such as edge/vertex deletion/addition and label change—to make the graphs isomorphic. For a graph $G$, we denote $\mathcal{N}(G)$ as the neighbor set of graphs that are only one edit distance away from $G$. We use the normalized GED distance $\widehat{\text{GED}}(G_1, G_2) = \text{GED}(G_1, G_2)/(|V_1| + |V_2| + |E_1| + |E_2|)$ for our framework, which has the advantage to compare two graphs of different sizes.

Since the GED is NP-hard to compute, we use a graph embedding algorithm, GREED from (Ranjan et al., 2023), as a proxy, which aims to learn a GED metric through a projection of the graph dataset in $\mathbb{R}^l$. Note that it is possible to employ any other embedding algorithm that can estimate the GED between two graphs, to be able to validate the closeness of counterfactuals, and to define common recourse on the space of graphs through vectors in $\mathbb{R}^l$. We denote as $z(G)$ the embedding of a graph $G$ in the rest of the paper.

### 3.2. Multi-head vertex reinforced random walk

To identify counterfactuals, we explore the graph edit map through a random walk. The random walk occurs in the space of graphs, where each state is a distinct (non-isomorphic) graph. Two states are connected if and only if their corresponding graphs can be transformed into one another by a single edit.

COMRECGC uses a variation of a vertex reinforced random walk (VRRW). VRRW (Pemantle, 2004) performs random walks on a finite state space where the transition probability depends on the number of visits. This family of random walks has the advantage of experimentally converging, as well as providing an interpretation of the diversity and exploration performances (Mei et al., 2010). We now describe how COMRECGC uses random walks:

Initially, the random walk begins with $k$-heads each placed on different input graphs randomly, $(G_i)_{i \leq k} \subset \mathbb{G}$. At each step, we either continue the walk or, with a small probability, all heads teleport back to starting graphs. It is crucial for our walk to keep track of the graph from which each head started or was teleported to. We represent the state of the $k$-heads random walk as $(u_i)_{i \leq k}$.

In each step, we randomly select one of the heads as the **lead**, denoted by the index $\ell$, then proceed as follows:

- First, we move the **lead head**. The goal for the walker is to go towards a potential counterfactual graph with the following transition rule, for $v \in \mathcal{N}(u_\ell)$:

$$p(u_\ell, v) \propto p_\phi(v)C(v) \qquad (2)$$

Where $C(v)$ is one plus the number of visits to graph $v$, and $p_\phi(v)$ is the probability, assigned by the GNN, of $v$ being a counterfactual.

- Each **non-lead head** moves to the next state based on the recourse available among its neighbors that is closest to the lead's recourse. More formally, if the lead head is in state $u_\ell$ after the previous step, then the next state for the $i$-th head is:

$$argmin_{v \in \mathcal{N}(u_i) \cup \{u_i\}} || \overrightarrow{z(G_\ell)z(u_\ell)} - \overrightarrow{z(G_i)z(v)} ||_2$$

**Teleportation.** The graph edit space is exponentially large. Thus, to explore the search space around small neighborhoods of the input graphs, we restrict the random walk by adding a probability of returning to input graphs of $0 < \tau < 1$ at each step. We call this operation teleportation, and each head state is reset to one of the input graphs optimizing for **coverage** as follows: define $t(G)$ to be the number of walks started from the input graph $G$. Then the probability of a head to teleport to $G \in \mathbb{G}$ is:

$$p_\tau(G) = \frac{\exp(-t(G))}{\Sigma_{G' \in \mathbb{G}} \exp(-t(G'))} \qquad (3)$$

*The pseudo code of the random walk procedure and some elements of analysis are presented in the Appendix (D.1 and D.2).*

**Counterfactual Candidates.** After $M$ steps, the random walk is terminated. For the next phase, we select the counterfactuals that have been visited at least a certain number of times, referring to these as the counterfactual candidates that represent the outcomes of the walk.

### 3.3. Aggregation with clustering

Once counterfactuals have been found, COMRECGC forms clusters among the resulting recourse to find common recourse. For this task, we use a spatial clustering algorithm, DBSCAN (Ester et al., 1996) to find high density areas to aggregate recourse with close embedding representation. We associate a common recourse to each cluster of radius

$\Delta$. Finally, we pick $R$ common recourse to be our explanation using a greedy approach. Algorithm 1 summarizes this method on a set $\mathbb{G}$ of input graphs, with a set of counterfactuals $\mathbb{S}$, where **gain**$(r; \mathbb{F})$ is the gain in coverage from adding the recourse $r$ to $\mathbb{F}$.

---
**Algorithm 1** CR CLUSTERING($\mathbb{G}$, $\mathbb{S}$, $R$)
---
1: $\mathbb{C} \leftarrow \Delta$-clusterize $\{\overrightarrow{z(G)z(v)} : v \in \mathbb{S}, G \in \mathbb{G}\}$
2: $\mathbb{F} \leftarrow \emptyset$
3: **for** $i \in 1 : R$ **do**
4: $\quad r \leftarrow \arg\max_{r \in \mathbb{C}}$ **gain**$(r; \mathbb{F})$
5: $\quad \mathbb{F} \leftarrow \mathbb{F} + \{r\}$
6: **end for**
7: **return** $\mathbb{F}$

---

### 3.4. COMRECGC: algorithm and complexity

Finally, in Algorithm 2 we present COMRECGC's solution to the FCR problem, which consists of a Multi-head VRRW, a selection of counterfactual candidates for recourse, and a clustering process for identifying common recourse. Line 2 is the filtering based on the number of visits, where $n$ is a threshold set as a hyperparameter for the dataset.

---
**Algorithm 2** COMRECGC($\phi$, $\mathbb{G}$, $k$, $M$, $\tau$, $R$, $n$)
---
1: $\mathbb{S} \leftarrow$ MULTI-HEAD VRRW($\phi$, $\mathbb{G}$, $k$, $M$, $\tau$)
2: $\mathbb{S} \leftarrow$ top $n$ frequently visited counterfactuals in $\mathbb{S}$
3: **return** CR CLUSTERING($\mathbb{G}$, $\mathbb{S}$, $R$)

---

**Complexity Analysis.** The complexity of the random walk is $O(Mhk)$, where $M$ is the number of steps in the VRRW, $k$ is the number of heads in the VRRW, and $h$ is the maximum node degree in the graph-edit space map. The complexity of using DBSCAN clustering on $n|\mathbb{G}|$ recourse is $O(n^2|\mathbb{G}|^2)$, where $n$ is the number of top-visited counterfactuals, and $|\mathbb{G}|$ is the number of input graphs. In practice, we have a small constant due to being in $\mathbb{R}^l$ for $l = 64$. Finally, the complexity of the greedy summary of $n|\mathbb{G}|$ recourse over $|\mathbb{G}|$ features $R$ times is $O(nR|\mathbb{G}|^2)$, where $R$ is the size of the recourse set. The overall complexity is $O(Mhk + n^2|\mathbb{G}|^2)$.

### 3.5. Variation of COMRECGC for the FC problem

Our goal is to build a generic framework to solve both the FCR and FC problems. We can also use COMRECGC to generate solutions to the FC problem. In the counterfactual candidates step, we need to significantly reduce the number of counterfactuals used to generate common recourse in order to match the constraints in FC. COMRECGC selects the counterfactuals that are closest to the input graphs. Intuitively, this is because the common recourse threshold $\Delta$ is geometrically tight, so we want recourse (cost) to be as small as possible to increase the chance of getting common

ones in the clustering step. Algorithm 3 represents COMRECGC's solution to the FC problem. Line 3 is specific to the FC framework. In particular, we select $T = |\mathbb{G}|$ counterfactuals. This allows our method to be compared to the existing local explainers, where one counterfactual is given per input graph (Ying et al.). The complexity of this extra step is $O(n|\mathbb{G}|)$, hence COMRECGC's solution to the FC problem has a complexity of $O(Mhk + n^2|\mathbb{G}|^2)$.

---
**Algorithm 3** COMRECGC for FC ($\phi$, $\mathbb{G}$, $k$, $M$, $R$)
---
1: $\mathbb{S} \leftarrow$ MULTI-HEAD VRRW($\phi$, $\mathbb{G}$, $k$, $M$)
2: $\mathbb{S} \leftarrow$ top $n$ frequently visited counterfactuals in $\mathbb{S}$
3: $\mathbb{S} \leftarrow \bigcup_{G \in \mathbb{G}} \arg\min_{v \in \mathbb{S}} ||\overrightarrow{z(G)z(v)}||_2$
4: **return** CR CLUSTERING($\mathbb{G}$, $\mathbb{S}$, $R$)

---

## 4. Experiments

We evaluate COMRECGC on four real-world datasets against the recourse from benchmark explainers, and show:

- COMRECGC produces global common recourse that are of higher quality than those of the baselines on both the FCR and FC problems.

- The explanations produced by COMRECGC are significantly less costly in terms of recourse than the ones generated by the baseline counterfactual explainers.

- The common recourse from COMRECGC can explain a similar, and in some cases higher, number of input graphs compared to the counterfactual graphs generated by a global counterfactual explainer.

*Reproducibility.* We make our code available at https://github.com/ssggreg/COMRECGC.

### 4.1. Datasets

We consider the datasets MUTAGENICITY (Riesen & Bunke, 2008; Kazius et al., 2005), NCI1 (Wale & Karypis, 2006), AIDS (Riesen & Bunke, 2008), and PROTEINS (Borgwardt et al., 2005; Dobson & Doig, 2003). In the first three, each graph accounts for a molecule, where nodes represent atoms and edges chemical bonds between them. The molecules are classified by whether they are mutagenic, anticancer, or active against HIV, respectively. The PROTEINS dataset is composed of different proteins classified into enzymes and non-enzymes, with nodes representing secondary structure elements and edges representing structural proximity. For each dataset, we remove graphs containing rare nodes (with a label count less than 50). The statistics of the datasets are detailed in Table 1.

*Table 1.* Datasets overview.

|  | NCI1 | MUTAGENICITY | AIDS | PROTEINS |
|---|---|---|---|---|
| #Graphs | 3978 | 4308 | 1837 | 1113 |
| #Nodes | 118714 | 130719 | 28905 | 43471 |
| #Edges | 128663 | 132707 | 29985 | 81044 |
| #Node Labels | 10 | 10 | 9 | 3 |

## 4.2. Experimental set up

We train a base GNN model (GCN) (Kipf & Welling, 2017) for a binary classification task, consisting of three convolutional layers, a max pooling layer, and a fully connected layer, following best practices from the literature (Vu & Thai, 2020). The model is trained with the Adam optimizer (Kingma & Ba, 2014) and a learning rate of 0.001 for 1000 epochs. The training/validation/testing split is 80%/10%/10%, and the corresponding accuracy measures are presented in Table 2.

*Table 2.* Accuracy of GCN on the graph binary classification task on NCI1, MUTAGENICITY, AIDS and PROTEINS.

|  | NCI1 | MUTAGENICITY | AIDS | PROTEINS |
|---|---|---|---|---|
| Training | 0.844 | 0.882 | 0.998 | 0.780 |
| Validation | 0.816 | 0.830 | 0.973 | 0.820 |
| Testing | 0.781 | 0.800 | 0.978 | 0.730 |

**Choice of the parameters $\theta$ and $\Delta$.** The parameter $\theta$ accounts for the distance to a graph $G \in \mathbb{G}$ within which a graph with an accepted prediction can be used as a counterfactual explanation for $G$. There is a trade-off: a small $\theta$ provides instance-wise explanations, if they exist, and a larger $\theta$ allows for more effective summaries, as counterfactuals tend to cover a broader range of input graphs. Following (Huang et al., 2023), we set $\theta = 0.1$ in the experiments on the NCI1, MUTAGENICITY and AIDS datasets. However, we set $\theta = 0.15$ for the PROTEINS dataset, as we have found experimentally that counterfactuals tend to be more distant from their input graphs.

On the other hand, the parameter $\Delta$ accounts for the maximum difference between two recourse to be considered common. A larger $\Delta$ yields better summaries but the insights become less precise. We set $\Delta = 0.02$, although this may seem large compared to the value of $\theta$, the embedding space dimension used for the datasets is $l = 64$, making the margin for common recourse sensibly smaller than the one for finding counterfactuals, as expected. An example illustrating a common recourse explanation with those parameters is provided in Figure 1.

**COMRECGC parameters.** Across all of our experiments, COMRECGC uses $k = 5$ heads, has probability of teleportation $\tau = 0.05$, performs the random walk for $M = 50000$ steps, and selects $R = 100$ common recourse.

## 4.3. Results for the FCR problem

**Baselines.** Since the FCR problem does not constrain the number of counterfactuals used to generate common recourse, explainers used for benchmarking this problem must be able to generate a large number of counterfactuals. In the literature, we find that only GCFEXPLAINER(Huang et al., 2023) aims at generating global counterfactuals on a large scale. This is also the only global method to generate counterfactual. To generate common recourse with GCFEX-PLAINER counterfactuals, we use the same recourse clustering algorithm (Algorithm 1) that is part of COMRECGC. For COMRECGC, its number of heads variants, and GCFEX-PLAINER, we limit the number of counterfactuals entering the clustering stage to $100,000$ graphs.

**Results.** The results are presented in Table 3. We observe that the common recourse explanation from COMRECGC yields better coverage by a significant margin on all datasets, thus providing a better solution for the FCR problem than the baselines. The cost of the common recourse is also lower on the NCI1, MUTAGENICITY datasets, comparable on the AIDS dataset, and slightly above on the PROTEINS dataset. This difference in coverage is explained by the multi-head random walk used to generate counterfactuals in COMRECGC, which tends to facilitate forming common recourse afterwards. The lower cost is also explained by the fact that finding common recourse with similarity threshold $\Delta = 0.02$ is difficult, and is easier on small recourse than on larger ones.

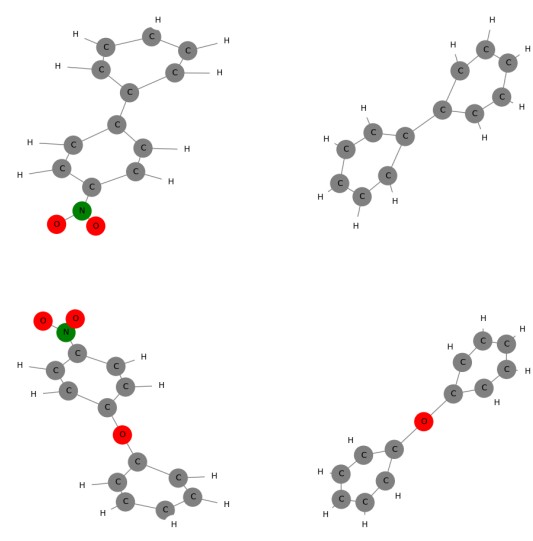

*Figure 1.* Common Recourse on MUTAGENICITY: Removing an $NO_2$ complex. On the left two mutagenetic molecules from the input, on the right two resulting non-mutagenetic molecules.

*Table 3.* Results on the FCR problem. Higher coverage corresponds to generating common recourse shared by more input graphs. The cost (lower is better) refers to the total length of the recourse. COMRECGC and its variants outperform the baseline in almost all settings.

| | NCI1 | | MUTAGENICITY | | AIDS | | PROTEINS | |
|---|---|---|---|---|---|---|---|---|
| | Coverage | Cost | Coverage | Cost | Coverage | Cost | Coverage | Cost |
| GCFEXPLAINER | 21.4% | 5.75 | 20.6% | 6.91 | 14.2% | 6.97 | 32.8% | **10.65** |
| COMRECGC 2-HEAD | 40.5% | 5.12 | 45.9% | 5.74 | 32.8% | 6.71 | 45.9% | 11.44 |
| COMRECGC 3-HEAD | 44.5% | 5.07 | **52.6%** | **5.61** | 33.6% | **6.62** | 45.9% | 11.51 |
| COMRECGC 4-HEAD | 44.6% | 4.70 | 52.0% | 5.81 | 34.8% | 6.71 | 46.2% | 11.47 |
| COMRECGC 5-HEAD | 42.9% | 4.95 | 51.8% | 5.63 | 34.7% | 6.74 | 46.4% | 11.59 |
| COMRECGC 6-HEAD | **44.9%** | **4.51** | 52.0% | 5.68 | **35.2%** | 6.66 | **47.3%** | 11.59 |

*Table 4.* Results on the FC problem. Higher coverage and lower cost are desirable. COMRECGC performs better than the baselines on all datasets and metrics except for the cost on the AIDS and PROTEINS datasets.

| | NCI1 | | MUTAGENICITY | | AIDS | | PROTEINS | |
|---|---|---|---|---|---|---|---|---|
| | Coverage | Cost | Coverage | Cost | Coverage | Cost | Coverage | Cost |
| DATASET COUNTERFACTUALS | 8.52% | 9.02 | 10.4% | 8.34 | 0.41% | **0.97** | 29.0% | 12.95 |
| LOCALRWEXPLAINER | 19.0% | 5.89 | 18.2% | 7.19 | 12.9% | 7.31 | 22.1% | 11.33 |
| GCFEXPLAINER | 14.7% | 7.12 | 11.9% | 7.80 | 14.2% | 7.07 | 29.8% | **11.13** |
| COMRECGC | **33.4%** | **5.60** | **46.7%** | **6.56** | **24.3%** | 6.59 | **39.6%** | 12.04 |

## 4.4. Results for the FC problem

**Baselines.** The FC problem constrains the number of counterfactuals used to form common recourse, which allows us to compare our method to common local counterfactual explainers. To evaluate the performance of our method compared to baselines, we construct a common recourse explanation for a counterfactual explainer as follows: a counterfactual explainer takes the set of input graphs $\mathbb{G}$ and returns a set $\mathbb{S}$ of counterfactuals of the same cardinality. We form clusters on $\mathbb{S}$ using Algorithm 1 to generate a common recourse explanation. The baseline counterfactual explainers are GCFEXPLAINER and a local random walk explainer. The second one performs a random walk around each input graph to find close counterfactuals; we select the closest counterfactual to each input graph to be part of the set of counterfactual candidates used to generate recourse. We have not used explainers such as CFF EXPLAINER (Bajaj et al., 2021), RCEXPLAINER (Tan et al., 2022), as they mainly focus on local explanation, and GCFEXPLAINER already outperforms these methods (Huang et al., 2023). They also only generate subgraph counterfactual explanations, whereas a random walk can also add elements to the graph to find counterfactuals. Finally, we also add a baseline that corresponds to Algorithm 1 processing the counterfactual graphs given in the original dataset.

**Results.** Table 4 presents the outcome. We show more detailed results for different numbers of recourse in Figure 2. We observe that COMRECGC generates the best coverage compared to all the baselines on the FC problem across all datasets. The cost of the common recourse is also lower than the recourse from other method, except on the Proteins dataset, where they are comparable. The low cost of the dataset counterfactuals common recourse for the AIDS dataset is explained by the low number of common recourse formed, as few $\theta$-close counterfactuals are in the dataset.

This difference in coverage is even more striking in the FC problem than in the FCR problem. When the number of counterfactuals is reduced from $100,000$ to $2000$, mostly close counterfactuals are grouped together; this is why the local RW explainer performances are better than the GCFEXPLAINER global counterfactual for forming common recourse on the NCI1 and MUTAGENICITY datasets.

## 4.5. Common recourse explanations vs global counterfactual explanations

Lastly, we compare common recourse explanations and global counterfactual explanations in terms of coverage. Since COMRECGC is the only method designed to explain through common recourse, we will compare the common recourse explanation generated by COMRECGC to the global graph counterfactuals generated by GCFEXPLAINER(Huang et al., 2023), CFF EXPLAINER (Bajaj et al., 2021), and RCEXPLAINER (Tan et al., 2022). We also add the global graph counterfactual explanation generated by the counterfactual given in the dataset.

For the definition of coverage for a set of counterfactual graphs $\mathbb{S}$, we refer to (Huang et al., 2023) and define it as the ratio of the input graphs covered by at least one counterfactual in $\mathbb{S}$. More formally $\text{COVERAGE}(\mathbb{S}) := \big|\{G \in$

*Table 5.* Common recourse explanation vs global counterfactual explanation for 10 explanations. COMRECGC's common recourse explanations outperform the baselines' graph counterfactual explanations on most of the datasets.

| | NCI1 | MUTAGENICITY | AIDS | PROTEINS |
| | Coverage | Coverage | Coverage | Coverage |
| --- | --- | --- | --- | --- |
| DATASET COUNTERFACTUALS | 16.5% | 29.0% | 0.4% | 8.5% |
| RCEXPLAINER | 15.2% | 32.0% | 9.0% | 8.7% |
| CFF | 17.6% | 30.4% | 3.4% | 3.8% |
| GCFEXPLAINER | **27.9%** | 37.1% | 14.7% | 10.9% |
| COMRECGC | 26.1% | **39.4%** | **15.2%** | **18.0%** |

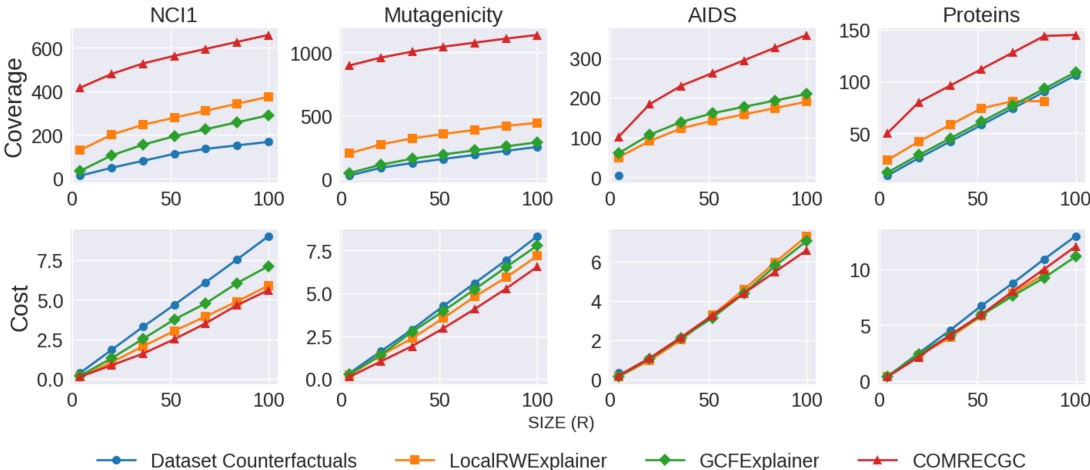

*Figure 2.* Common Recourse coverage and cost comparison between COMRECGC and baselines for the FC problem where $\Delta = 0.02, T = |\mathbb{G}|$ and $R = 1$ to $100$ common recourse.

$\mathbb{G}|min_{S \in \mathbb{S}}||z(G) - z(S)||_2 \leq \theta\}|/|\mathbb{G}|$.

The results, shown in table 5 for 10 explanations, indicate that the common recourse explanations from COM-RECGC are comparable to the best graph counterfactual method, GCFEXPLAINER, on the NCI1, MUTAGENICITY, and AIDS datasets, while being significantly more comprehensive on the PROTEINS dataset. This difference may be due to the sparsity of the Proteins dataset, which makes it challenging to identify a central counterfactual graph to explain the sparse neighborhood. Common recourse seems to be a more suitable explanation for this dataset, at least under the conditions of our experiments.

### 4.6. Experiments on different values of $\theta, \Delta$

**Setting.** We experiment on different values of $\theta, \Delta$. Those parameters are only used in the clustering algorithm part of COMRECGC. The results are presented in Table 6, for COMRECGC with parameters $M = 50,000, k = 5, \tau = 0.05$.

**Results.** We do not run into issues with increasing the value of $\Delta$, the common recourse threshold. As expected, the coverage goes up, and so does the cost as COMRECGC is

able to cover close to the whole of MUTAGENICITY and NCI1 datasets, and follows the same trend for the AIDS dataset. However, it is quite surprising that the coverage on the PROTEINS dataset went down by $1\%$. An explanation for this is a possible issue with the clustering algorithm DBScan, which may be caused by a too big overlap on some common recourse clusters.

Increasing the counterfactual threshold $\theta$ is proven to be challenging, as raising it from $0.1$ to $0.15$ across datasets dramatically increased the number of recourse entering the clustering stage. For example, on the MUTAGENICITY and NCI1 datasets, the number of recourse increased by a factor of $100$, making DBScan intractable in its current form. As expected, increasing $\theta$ results in greater coverage in the common recourse explanations produced by COMRECGC, but it also raises the cost, as longer recourse are allowed. In particular, on the AIDS dataset, the increase in coverage was minimal, suggesting a potential bottleneck in the effectiveness of common recourse explanations for that dataset with common recourse threshold set at $\Delta = 0.02$.

In Table 7 we experiment with a finer variation of $\Delta$ and its effect on the coverage variation. We observe that a higher $\Delta$

*Table 6.* Results on the FCR problem of COMRECGC for different values of $\theta$ and $\Delta$. We find that increasing the common recourse and counterfactual threshold tends to improve coverage, particularly for more dense datasets, such as MUTAGENICITY and NCI1. The effect on more sparse datasets such as PROTEINS and AIDS is more contrasted, with moderate to no added coverage. The cost rises as we allow more distant counterfactuals. For configurations where $\theta = 0.15$ and $\Delta = 0.02$, the number of recourse entering the clustering stage is large, making the DBScan clustering algorithm intractable, resulting in missing values ('x').

| | NCI1 | | MUTAGENICITY | | AIDS | | PROTEINS | |
|---|---|---|---|---|---|---|---|---|
| | Coverage | Cost | Coverage | Cost | Coverage | Cost | Coverage | Cost |
| $\theta = 0.1, \Delta = 0.02$ | 42.9% | 4.95 | 51.8% | 5.63 | 34.7% | 6.74 | 42.8% | 7.21 |
| $\theta = 0.1, \Delta = 0.04$ | 86.5% | 8.05 | 90.0% | 8.11 | 47.0% | 8.4 | 41.5% | 6.36 |
| $\theta = 0.15, \Delta = 0.02$ | x | x | x | x | 34.8% | 7.65 | 46.4% | 11.59 |

*Table 7.* Results on the FCR problem of COMRECGC for different values of $\theta$ and $\Delta$ on the Mutagenicity dataset.

| | MUTAGENICITY | |
|---|---|---|
| | Coverage | Cost |
| $\theta = 0.1, \Delta = 0.02$ | 51.8% | 5.63 |
| $\theta = 0.1, \Delta = 0.025$ | 67.15% | 6.99 |
| $\theta = 0.1, \Delta = 0.03$ | 75.35% | 7.59 |
| $\theta = 0.1, \Delta = 0.035$ | 83.88% | 8.03 |
| $\theta = 0.1, \Delta = 0.04$ | 90.0% | 8.11 |

allows for a more comprehensive explanation, as we allow similar recourse to be considered "common" more easily. The cost naturally rises as more explanations are covered.

### 4.7. Discussions

We make a few important observations, including a few limitations of COMRECGC. Thee observations will be helpful in practice. *First,* since the FCR and FC problems are novel, our method is the only one specifically designed to generate common recourse. It is therefore unsurprising that we outperform the existing counterfactual explainers. *Second,* determining the appropriate parameters $\theta$ and $\Delta$ for the FCR problem is challenging and application dependent. Since these parameters serve as thresholds for a normalized distance, larger graphs generally work well with smaller values of $\theta$ and $\Delta$, while smaller graphs require larger values. *Third,* we believe that relaxing the values of $\theta$ and $\Delta$ could lead to richer and more diverse explanations. However, as more common recourse would be generated in that process, we need an appropriate filtering system to select the most meaningful ones. *Fourth,* we explore the graph edit map without considering the physical feasibility of the changes, especially if the domain has specific constraints (e.g., chemical bonds in molecules). As a result, the generated counterfactual graphs may not correspond to the real-world entities depending on the domain.

*We discuss the choice of the random walk parameters in Appendix E.1. We present running times of our method in Appendix E.2, an ablation study in E.3, and other examples of common recourse in E.4. In Appendix E.5 we present results on the different types of GNNs. In Appendix E.6 we test our method on a network dataset and add a comparison of our method to a local CE explainer in Appendix E.7.*

## 5. Conclusion

In this work, we have formalized the problem of generating global counterfactual explanations for GNNs with common recourse. This novel problem setting includes the FCR and FC problems. These problems are NP-hard and thus, we have designed COMRECGC, a method specifically tailored to extract high-quality common recourse explanations. Through extensive experiments on real-world datasets, we have benchmarked COMRECGC against popular counterfactual explainers in the common recourse task. Our results demonstrate that COMRECGC produces global common recourse of significantly higher quality than the baselines across both the FCR and FC problems. Finally, COMRECGC's common recourse explanations can account for a similar number of input graphs as those generated by global counterfactual explainers, providing a robust and scalable alternative to global graph counterfactual explanation.

## Acknowledgment

We would like to thank György Turán for his generous support, patience, and care throughout this project, and for initiating this collaboration. We also thank Sayan Ranu for the helpful discussions. The work has been supported in part by NSF grant ECCS-2217023. The authors also acknowledge the National Artificial Intelligence Research Resource (NAIRR) Pilot and the Texas Advanced Computing Center (TACC) Vista for contributing to this research result.

## Impact Statement

This paper presents work to advance the field of explainable Machine Learning. There are many potential societal consequences of our work, none which we feel must be specifically highlighted here.

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

# Appendix

## A. Additional Related Work

We summarize recent approaches for counterfactual explanations. (Abrate et al.) propose counterfactual explanations by sparsifying and densifying graphs, to find counterfactuals through pattern deletion or generation. ARES (Rawal & Lakkaraju, 2020) produces actionable two-level recourse summaries on non-graphical datasets through an optimization problem. Lastly, in (Magister et al., 2021), the "Human-in-the-Loop" framework integrates human feedback to enhance the relevance and interpretability of counterfactual explanations. Recent advancements include (He et al., 2024) generative flow network, (Qiu et al., 2024) robust counterfactual witnesses,(Kang et al., 2024) focus on unsupervised node representation learning.

## B. Example for the FC problem

**Setup.** We follow the setting described below.

- Given a set of input graphs: $\mathbb{G} = \{G_1, G_2, G_3\}$.

- And counterfactual graphs: $\mathbb{H} = \{H_1, H_2, H_3, H_4\}$.

- Each counterfactual $H$ is obtained by applying a recourse $f$, i.e., a graph transformation, to an input graph $G$ according to the following table:

**Finding Counterfactuals to Maximize Common Recourse Coverage.** Suppose the recourse transforms the input graph as follows:

| Input Graphs | Recourse | Counterfactual Graphs |
|:---:|:---:|:---:|
| $G_1$ | $f_1$ | $H_1$ |
| $G_1$ | $f_2$ | $H_2$ |
| $G_2$ | $f_3$ | $H_3$ |
| $G_3$ | $f_1$ | $H_4$ |
| $G_3$ | $f_3$ | $H_1$ |

We will write $f_1(G_1) = H_1$ to denote that the recourse $f_1$ applied to the input graph $G_1$ yields the counterfactual graph $H_1$. We say that an input graph $G \in \mathbb{G}$ is *covered* by recourse $f$ if:

(i) both $H$ and $f$ have been chosen within the budget, and

(ii) $f(G) = H$.

Given budgets $R = 2$ (recourse) and $T = 2$ (counterfactuals), the goal of the FC problem is to select 2 counterfactuals in $\mathbb{H}$ that yield the best coverage on $\mathbb{G}$ using at most 2 recourse.

**Application.** Suppose we choose counterfactual graphs $H_1$ and $H_3$, then the best two recourse to pick are: $\mathbb{F}_{\mathbb{H}^*} = \{f_1, f_3\}$, which allows us to cover the three input graphs as follows:

$$f_1(G_1) = H_1, \quad f_1(G_3) = H_1, \quad f_3(G_2) = H_3.$$

An intuitive way to see this problem is as a *"max 2-budget 2-cover problem"*, where 2-cover means that to "cover" an element, one has to cover it in the two budgets.

## C. Additional Proofs and Analyses for Section 2

### C.1. Proof of Theorem 1

*Proof.* We prove that the FCR problem is NP-Hard through a reduction from the maximum coverage problem. Consider the problem of covering $U$ using the sets $(S_i)_{i \leq m}$, we build the following instance of the FCR problem:

Let $X$ be a graph binary classifier that accepts a graph if and only if the graph has two nodes of the same color. To each $u_j \in U$, we associate $G_j$, a star graph with an unlabelled central node and $m$ peripheral vertices. If $u_j$ is in $k$ elements of $S$, we one-color $k$ peripheral vertices with the colors $\{i : u_j \in S_i\}$. We define the common recourse $r_i$, for $i \leq m$, as follows: $r_i$ colors the middle vertex of a star graph using color $i$, and we take the corresponding counterfactuals as the inputs to this FCR problem.

Thus $X(G_j) = 0$ for all $j$ and $X(r_i(G_j)) = 1$ if and only if $u_j \in S_i$. Hence from an optimal solution to this FCR problem, we derive an optimal solution for the maximum coverage problem $(U, (S_i)_i)$. Therefore FCR is NP-hard. $\square$

## C.2. Analysis for The FCR problem

Let us define $f$ as the function that associates to a set of common recourse its total coverage. It is not hard to see that $f$ is monotone, i.e $f(\mathbb{F} \cup \{r\}) \geq f(\mathbb{F})$ for every set of recourse $\mathbb{F}$, and recourse $r$. We recall the definition of submodularity (Santiago & Yoshida, 2020), a function $h$ is submodular if for any two sets $A \subseteq B$ and for any element $e$:

$$h(A \cup \{e\}) - h(A) \geq h(B \cup \{e\}) - h(B) \tag{4}$$

In the rest of the paper, we denote by $h_A(B)$ the marginal gain of adding the set $B$ to $A$, i.e $h(A \cup B) - h(A)$. We now prove that $f$ is submodular:

*Proof.* Let $A \subseteq B$ be sets of recourse, and let $r$ be a recourse. Suppose $G$ is a counterfactual graph covered by $r$ but no the any recourse in $B$, since $A \subseteq B$, $r$ is not covered by any elements of $A$. Therefore $f_A(\{r\}) \geq f_B(\{r\})$. $\square$

## C.3. Budget Version of the FC problem

When we assume that the counterfactuals and recourse are all known, the FC problem becomes a budget problem:

**Problem** 3 (max 2-budget 2-cover problem). Given two budgets $k_1$ and $k_2$, and two families of sets $\mathbb{S}_1, \mathbb{S}_2$, the goal is to find $\mathcal{S}_1 \subset \mathbb{S}_1$, $\mathcal{S}_2 \subset \mathbb{S}_2$ verifying :

$$max_{\mathcal{S}_1, \mathcal{S}_2} S_1 \cap S_2 \text{ st. } (\mathcal{S}_1) \leq k_1 \text{ and } size(\mathcal{S}_2) \leq k_2,$$

where $S_i = \bigcup_{S \in \mathcal{S}_i} S$.

To the best of our knowledge, this problem has not been studied. This is a variation of the maximum coverage problem. The budgeted maximum coverage problem (Khuller et al., 1999) is another budget variation, where only one budget is considered, there is no constraint about the intersection, but the sets costs and the rewards for covering elements can be different from 1. The multi-budget maximum coverage problem is another variation between knapsack and maximum coverage (Cellinese et al., 2021), but there is only one family of sets to pick from, and we do not require double coverage. Another variation of the maximum coverage problem is called multi-coverage (Barman et al., 2019), where covering multiple times one element raises the objective value.

Another variation is called multi-set multi-cover, which uses one budget, and each element must be covered a certain number of times (Hall & Hochbaum, 1986; Hua et al., 2009; Gorgi et al., 2021).

## C.4. Proof of Theorem 2

A function $h$ is called pseudo-modular if there exists $\gamma \in (0, 1]$ such that for any pair of disjoint sets $A, B$:

$$\sum_{e \in B} h_A(e) \geq \gamma h_A(B) \tag{5}$$

Note that this $\gamma$ is the minimum over all the $\gamma_{A,B}$, for all sets $A, B$, where $\gamma_{A,B}$ is a specific value of $\gamma$ for given $A, B$ in equation 5. We now proceed to the proof of Theorem 2:

*Proof.* Recall $g$ is the function that associates a set of counterfactuals with its best coverage through building common recourse.

We give an instance of the FC problem, and two sets $A, B$ such that equation 5 is only true for $\gamma_{A,B} = 0$, which means that there is no local pseudo-submodularity bound. Consider the instance of the FC problem where $R = 2$, $T \geq 4$ and $(G_i)_{1 \leq i \leq 4}$ some counterfactual graphs such that they each cover a different input graphs, $G_1$ is part of $R_1$ (recourse 1), $G_2$ part of $R_2$ (recourse 2), $G_3, G_4$ parts of $R_3$ (recourse 3). We set $A$ to $\{G_1, G_2\}$ and $B$ to $\{G_3, G_4\}$. Then $g_A(G_3) = g_A(G_4) = 0$ and $g_A(B) = 1$. $\square$

## C.5. Proof of Theorem 3

We first prove that $g$, the optimization function for the FC problem with the added constraint **C1**, is pseudo-modular.

*Proof.* For every set $A$, and for every graph $e$ in the problem space we have that $R \geq g_A(e)$ by problem definition, we also have that $g_A(e) > 0$ by **C1**. So for two disjoint sets $A, B$ part of the solution :

$$\sum_{e \in B} g_A(e) \geq \frac{1}{R|B|} g_A(B).$$

Hence, $g$ is pseudo-modular. □

The pseudo code of the Greedy algorithm for the FC problem with **C1** is as follows, where $g$ denotes the function that associates to a set of counterfactual its best coverage through building common recourse and selecting the $R$ best for coverage, and $T$ is the maximum number of counterfactuals to use for the explanations. This algorithm was first introduced

---

**Algorithm 4** GREEDY FC$(g, T)$

---

1: Initialize the set of counterfactuals: $S_0 \leftarrow \emptyset$
2: **for** $i \in 1 : T$ **do**
3:     Let $M_i \subseteq E \setminus S_{i-1}$ be a subset of size $T$ maximizing $\sum_{e \in M_i} g_{S_{i-1}}(e)$
4:     Let $e_i$ be an element uniformly chosen at random of $M_i$
5:     $S_i \leftarrow S_{i-1} + e_i$
6: **end for**
7: **return** $S_T$

---

in the work of (Buchbinder et al., 2014). Applying Theorem 1.10 of (Santiago & Yoshida, 2020) to the monotone and pseudo-modular function $g$ (Theorem 3), shows that Algorithm 4 yields a $1 - e^{-1/R}$ approximation in expectation to the FC problem, where $R$ is the maximum number of common recourse of the FC problem.

# D. Detail of our method, COMRECGC for Section 3

## D.1. The Pseudocode of MULTI-HEAD VRRW

The pseudo code for our multi-head VRRW is presented in Algorithm 5, where $G_i$ denotes the input graph where the random walk started or was last teleported for the $i$-th head, and $H_i$ denotes the current graph head $i$ is occupying: Lines 1-2 describe the initialization of the random walk, where $\mathbb{S}$ represents the set of counterfactuals we aim to find. Lines 5-12 outline the regular behavior of the random walk as it moves towards counterfactuals. Lines 13-15 cover the teleportation process. Finally, lines 17-21 detail the update of the counterfactual set. In line 23, we advance one step in the random walk, and in line 24, we update the visit count of the graphs reached by the $k$ heads during this step, following the principles of VRRW. This visit count influences the transition probabilities in Equation 2.

## D.2. Analysis of the Random Walks

**VRRW.** A vertex reinforced random walk is a random process with reinforcement that focuses on the number of visits of vertices (Pemantle, 2004) (Pemantle, 2007). To the best of our knowledge, theorical guarantees have only been derived for the special case where the initial transition matrix is symmetric (Volkov, 2001) (Benaïm & Tarrès, 2011), which is not relevant for our application, as we want to identify counterfactuals with the random walk, as described in Equation 2.

**RW.** Let us consider a classic, non-reinforced, random walk with the following transition rule:

$$p(u_\ell, v) \propto p_\phi(v) \tag{6}$$

Where $p_\phi(v)$ is the probability, assigned by the GNN, of vertex $v$ being a counterfactual, and uniform teleportation:

$$p_\tau(G) = \frac{1}{|\mathbb{G}|} \tag{7}$$

We refer to (Lovász, 1993) for the theoretical results. The main purpose of the analysis for our application is to determine the mixing time and mixing rate of our walk, to tune the parameter $M$, the number of steps. We add to the random walk the

**Algorithm 5** MULTI-HEAD VRRW($\phi, \mathbb{G}, k, M, \tau$)

1: $(G_1, \ldots, G_k) \leftarrow$ random input graph from $\mathbb{G}, \mathbb{S} = \emptyset$
2: $(H_1, \ldots, H_k) \leftarrow (G_1, \ldots, G_k)$
3: **for** $t \in 1 : M$ **do**
4:     Let $\epsilon, \ell \sim Bernoulli(\tau), \mathcal{U}\{1, \ldots k\}$
5:     **if** $\epsilon = 0$ **then**
6:         **for** $v \in \mathcal{N}(H_\ell)$ **do**
7:             Compute $P(H_\ell, v)$ based on equation 2.
8:         **end for**
9:         $v_\ell \leftarrow$ random neighbor of $H_\ell$ based on $p(H_\ell, v)$
10:         **for** $i \in 1 : k, i \neq \ell$ **do**
11:             $v_j \leftarrow argmin_{v \in \mathcal{N}(u_i)} || \overrightarrow{z(G_\ell)z(u_\ell)} - \overrightarrow{z(G_i)z(v)} ||_2$
12:         **end for**
13:     **else**
14:         $(G_1, \ldots, G_k) \leftarrow$ random input graphs from $\mathbb{G}$ based on equation 3.
15:         $(v_1, \ldots, v_k) \leftarrow (G_1, \ldots, G_k)$.
16:     **end if**
17:     **for** $i \in 1 : k$ **do**
18:         $C(v_i) \leftarrow C(v) + 1$
19:         **if** $\phi(v_i) > 0.5$ **then**
20:             $\mathbb{S} \leftarrow \mathbb{S} + \{v_i\}$
21:         **end if**
22:     **end for**
23:     $(H_1, \ldots, H_k) \leftarrow (v_1, \ldots, v_k)$.
24:     $N(H_1) = N(H_1) + 1, \ldots, N(H_k) = N(H_k) + 1$.
25: **end for**
26: **return** $\mathbb{S}$

following constraint: the maximum distance of the head from its starting point is at most $3\theta/2$, to make the space of the random walk finite. By Corollary 5.2 in (Lovász, 1993), the mixing rate of the walk is $max(|\lambda_2|, |\lambda_n|)$, where $\lambda_i$ is the $i$-th largest eigenvalue of the matrix $D^{-1/2}M_{rw}D^{1/2}$, where $M_{rw}$ is the transition matrix associated to our random walk and $D$ is the diagonal matrix with value $1/degree(i)$.

Unfortunately, there are two main obstacles that make this approach untractable: **(1)** The search space, the number of nodes of the graph on which we do the random walk, is of size $O(|\mathbb{G}|e^r\theta)$, hence making it hard to extract the eigenvalues of the matrix $M_{rw}$ **(2)** Each node's transition rule is determined by the GNN's prediction, making the construction of the matrix $M_{rw}$ difficult.

# E. Additional Details on Experiments & Results

### E.1. Choice of the random walk parameters $k, M, \tau$

The values of parameters $k$ and $\tau$ have been chosen through experiments on the Mutagenicity datasets. We chose $\tau$, the probability of teleportation, to be a compromise between the random walk being able to look far enough to find counterfactuals, while still being able to explore a good number of paths. Although higher values of $k$ seem to yield slightly better coverage, as seen in Table 3, $k$ increases the running time, the number of steps for the random walk to converge, and the memory usage. Hence, we decide to limit $k$ to 5 heads and to set $M$ to a relatively low $50,000$ steps.

### E.2. Running time

We show the running time of COMRECGC, its VRRW (Algorithm 5) and Clustering (Algorithm 1) components, for the FCR problem in the datasets of our study in Table 8. The running time follows the complexity given in Section 3, with shorter running times for small datasets such as AIDS compared to Mutagenicity. The sparsity and size of the graphs in the datasets also play a role, allowing the walk to explore more graphs in dataset such as Proteins.

*Table 8.* The running time in minutes of the different parts of COMRECGC for the experiments of Table 3.

|  | NCI1 | MUTAGENICITY | AIDS | PROTEINS |
|---|---|---|---|---|
| Algorithm 5 | 101 | 172 | 61 | 152 |
| Algorithm 1 | 23 | 27 | 14 | 14 |
| COMRECGC | 124 | 199 | 75 | 166 |

## E.3. Ablation Study

We give the performance of the variation of COMRECGC without some of its features on the FCR problem for the datasets in our study. We find the clustering to be very important on large datasets such as MUTAGENICITY, but less so on the smaller PROTEINS. This is possibly explained by the ability of the VRRW to spend a longer time pairing the same elements through different recourse, hence somewhat harmonizing some recourse.

*Table 9.* Results on the FCR problem of different variations of COMRECGC, in the same parameters setting as Section 4.

|  | NCI1 | | MUTAGENICITY | | AIDS | | PROTEINS | |
|---|---|---|---|---|---|---|---|---|
|  | Coverage | Cost | Coverage | Cost | Coverage | Cost | Coverage | Cost |
| COMRECGC | **42.9**% | 4.95 | **51.8**% | 5.63 | **34.7**% | 6.74 | **46.4**% | 11.59 |
| COMRECGC w/o clustering | 10.1% | x | 8.2% | x | 13.6% | x | 33.9% | x |
| COMRECGC w/o multi-head | 21.4% | 5.75 | 20.6% | 6.91 | 14.2% | 6.97 | 32.8% | **10.65** |

## E.4. Additional examples of common recourse

We provide additional examples of common recourse identified through COMRECGC, in the settings $\theta = 0.1$ and $\Delta = 0.02.$, in Figures 3 and 4.

In Figure 3, we observe larger molecules, which allow for a 'larger recourse' consisting of three transformations. This is because transformations on larger molecules correspond to smaller variations in the embedding for the "same transformation", which is capped by $\Delta$.

In Figure 4, on smaller molecules, we observe two types of transformations in the recourse, likely due to the increased value of the $\theta$ parameter compared to Figure 1. Indeed, a larger $\theta$ allows for more distant counterfactuals.

## E.5. Experiments with GAT, SAGE, GIN types of GNN

We train GAT(Velickovic et al., 2018), GraphSAGE(Hamilton et al., 2017), GIN (Xu et al.) GNNs model for a binary classification task, consisting of three convolutional layers, a max pooling layer, and a fully connected layer, following best practices from the literature (Vu & Thai, 2020). The model is trained with the Adam optimizer (Kingma & Ba, 2014) and a learning rate of 0.001 for 1000 epochs. The training/validation/testing split is 80%/10%/10%, and the corresponding accuracy measures are presented in Tables 10, 11, and 12.

*Table 10.* Accuracy of GAT on the graph binary classification task on NCI1, MUTAGENICITY, AIDS and PROTEINS.

|  | NCI1 | MUTAGENICITY | AIDS | PROTEINS |
|---|---|---|---|---|
| Training | 0.788 | 0.845 | 0.991 | 0.780 |
| Validation | 0.741 | 0.802 | 0.973 | 0.820 |
| Testing | 0.748 | 0.781 | 0.973 | 0.730 |

We experiment on COMRECGC and get the results for the FCR problem in Tables 13, 14 and 15. The parameters for COMRECGC are the same as for the experiments in Section 4 and Table 3, in particular $\theta = 0.1$ and $\Delta = 0.02$.

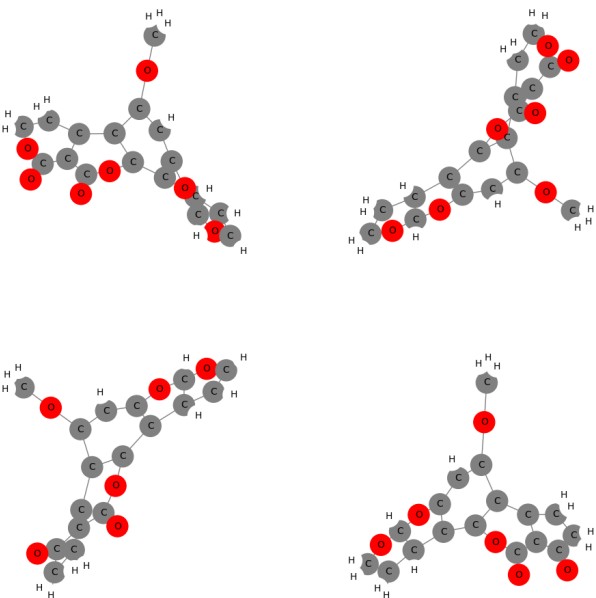

*Figure 3.* Common Recourse on the MUTAGENICITY dataset: removing two Hydrogen and one Carbon, on the left two mutagenetic input graphs, on the right two non-mutagenetic graphs.

*Table 11.* Accuracy of GraphSAGE on the graph binary classification task on NCI1, MUTAGENICITY, AIDS and PROTEINS.

|  | NCI1 | MUTAGENICITY | AIDS | PROTEINS |
|---|---|---|---|---|
| Training | 0.854 | 0.896 | 0.992 | 0.825 |
| Validation | 0.783 | 0.856 | 0.978 | 0.847 |
| Testing | 0.809 | 0.795 | 0.940 | 0.748 |

### E.6. Additional Dataset

.

We study the MDB-BINARY and MDB-MULTI datasets (Yanardag & Vishwanathan, 2015). The MDB-BINARY features movies from two genres (Action and Romance), where each graph represents a co-occurrence network of actors in a movie. A recourse represents a way to change the prediction from an action movie to a romance movie. The IMDB-MULTI includes movies from three genres (Comedy, Romance, and Sci-Fi). Since our current method is only interested in binary classification, we consider the following labels: Comedy and non-Comedy (i.e, Romance and Sci-fi). A recourse represents a way to change the prediction from a comedy movie to a non-comedy movie.

The parameters for COMRECGC are the same as for the experiments in Section 4 and Table 3, in particular $\Theta = 0.1$ and $\Delta = 0.02$. The results are presented in Table 16.

### E.7. Comparison to local CE explainer

.

We compare our method and GCFE to the popular local counterfactual baseline CF-GNNExplainer (Lucic et al., 2022) in Table E.7.

The coverage of building common recourse explanations using only CF-GNNExplainer-generated counterfactuals is noticeably worse than the counterfactual mining methods, such as ours or GCFExplainer. This is likely explained by a number of reasons, such as (i) generating fewer counterfactual graphs (around 2k5 graphs for CF-GNNexplainer on Mutagenicity vs 50k+ for GCFExplainer and our method), (ii) only considering subgraphs of input graphs and (iii) not taking into account the $\theta$ and $\Delta$ parameters of the CR explanation.

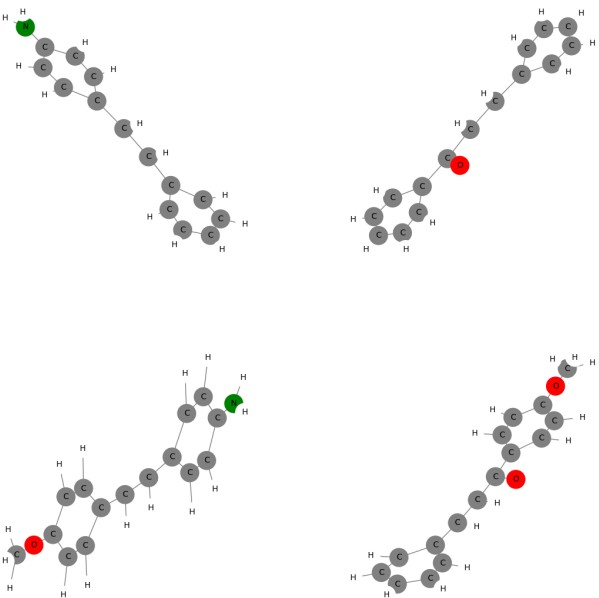

*Figure 4.* Common Recourse on the MUTAGENICITY dataset: removing one Nitrogen, one Hydrogen, adding one Oxygen and one Carbon, on the left two mutagenetic input graphs, on the right two non-mutagenetic graphs.

*Table 12.* Accuracy of GIN on the graph binary classification task on NCI1, MUTAGENICITY, AIDS and PROTEINS.

|  | NCI1 | MUTAGENICITY | AIDS | PROTEINS |
|---|---|---|---|---|
| Training | 0.863 | 0.849 | 0.999 | 0.810 |
| Validation | 0.826 | 0.800 | 0.951 | 0.847 |
| Testing | 0.789 | 0.784 | 0.946 | 0.748 |

*Table 13.* Results on the FCR problem for the task of explaining the GAT trained model. The parameters for recourse are the same as for the experiments in Section 4 and Table 3, in particular $\theta = 0.1$ and $\Delta = 0.02$.

|  | NCI1 | | MUTAGENICITY | | AIDS | | PROTEINS | |
|---|---|---|---|---|---|---|---|---|
|  | Coverage | Cost | Coverage | Cost | Coverage | Cost | Coverage | Cost |
| GCFEXPLAINER | 24.4% | 5.26 | 47.3% | **5.82** | 27.6% | 7.12 | 42.6% | 10.54 |
| COMRECGC (Ours) | **35.6%** | **5.02** | **55.7%** | 6.05 | **30.7%** | **6.89** | **42.9%** | **10.27** |

*Table 14.* Results on the FCR problem for the task of explaining the GraphSAGE trained model. The parameters for recourse are the same as for the experiments in Section 4 and Table 3, in particular $\theta = 0.1$ and $\Delta = 0.02$

|  | NCI1 | | MUTAGENICITY | | AIDS | | PROTEINS | |
|---|---|---|---|---|---|---|---|---|
|  | Coverage | Cost | Coverage | Cost | Coverage | Cost | Coverage | Cost |
| GCFEXPLAINER | 32.8% | 4.86 | 46.5% | **5.46** | 20.3% | 7.38 | 68.6% | 11.53 |
| COMRECGC (Ours) | **47.9%** | **4.76** | **50.9%** | 5.90 | **21.5%** | **7.16** | **69.4%** | **11.51** |

*Table 15.* Results on the FCR problem for the task of explaining the GIN trained model. The parameters for recourse are the same as for the experiments in Section 4 and Table 3, in particular $\theta = 0.1$ and $\Delta = 0.02$

|  | NCI1 | | MUTAGENICITY | | AIDS | | PROTEINS | |
|---|---|---|---|---|---|---|---|---|
|  | Coverage | Cost | Coverage | Cost | Coverage | Cost | Coverage | Cost |
| GCFEXPLAINER | 31.2% | 5.13 | 30.4% | **6.05** | 14.7% | 7.68 | 47.3% | 12.21 |
| COMRECGC (Ours) | **45.6%** | **4.58** | **33.7%** | 6.41 | **16.6%** | **7.34** | **48.6%** | **11.32** |

*Table 16.* Results on the FCR problem for the task of explaining the GIN trained model. The parameters for recourse are the same as for the experiments in Section 4 and Table 3, in particular $\theta = 0.1$ and $\Delta = 0.02$.

| | **IMDB-BINARY** (Yanardag & Vishwanathan, 2015) | | **IMDB-MULTI**(Yanardag & Vishwanathan, 2015) | |
|---|---|---|---|---|
| | Coverage | Cost | Coverage | Cost |
| GCFEXPLAINER | 76.5% | 8.33 | 19.9% | **7.65** |
| COMRECGC | **80.9%** | **8.10** | **21.9%** | 7.70 |

*Table 17.* Results on the FCR problem. Higher coverage corresponds to generating common recourse that is shared by more input graphs. The cost (lower is better) refers to the total length of the recourse.

| | NCI1 | | MUTAGENICITY | | AIDS | | PROTEINS | |
|---|---|---|---|---|---|---|---|---|
| | Coverage | Cost | Coverage | Cost | Coverage | Cost | Coverage | Cost |
| CF-GNNEXPLAINER | 9.1% | 7.5 | 8.4% | 8.90 | 0.1% | **0.35** | 0% | **0** |
| GCFEXPLAINER | 21.4% | 5.75 | 20.6% | 6.91 | 14.2% | 6.97 | 32.8% | **10.65** |
| COMRECGC (OURS) | **42.9%** | **4.95** | **51.8%** | **5.63** | **34.7%** | 6.74 | **46.4%** | 11.59 |

