# OpenReview forum: "COMRECGC: Global Graph Counterfactual Explainer through Common Recourse"
_ICML.cc/2025/Conference — ICML 2025 poster_

### Official Review · Reviewer_oYnL · 2025-03-05

**Overall Recommendation:** 3

**Summary:**

The authors focus on the question of generating global counterfactual explanations. They introduce the problem from a theoretical perspective. They introduce the problem of FCR and FC which correspond to Finding Common Recourse and Finding Counterfactual problems, respectively. They introduce these problems from a high-level theoretical perspective. They provide several theoretical guarantees such as: proving that FCR is a NP-hard problem, and that there exists an approximation for the FC problem with the given constraints. This motivates their algorithm, which uses a graph embedding, a multi-head vertex reinforced random walk to find CFs, and finally clustering to obtain common recourses.
After designing their algorithm they implement it on several real-world datasets. They use the data sets MUTAGENICITY, NCI1, AIDS, PROTEINs. In particular, most of these datasets are in molecules. They train a base GNN (GCN) and experiment with parameters $\theta$ and $\Delta$. They then compute the coverage with respect to the cost.

**Claims And Evidence:**

S1. The paper is well written and well founded. The main idea is novel and uses theoretical justification to motivate their framework.
S2. Their reasoning behind design choices is mostly sound.

S3. They provide initial experiments that showcase some of the promising behavior of their framework.

O1. Their is a glaring lack of experiments. It is understandable the authors introduce a novel problem and emphasize that in their experiments however the main metrics are coverage and cost. These experiments are good at showing their methods desired properties but they do not showcase mainstream metrics and experiments such as validity, sparsity, and potentially some other metrics. All these metrics are not necessary but they bolster their framework's strengths.

O2. Another issue is the lack of experiments. There are numerous types of datasets that have not been experimented on. Focusing primarily on molecular data leaves a glaring question for the readers: experimentally how does this framework work on other types of graphs (such as networks, etc.)? Not addressing this question also raises questions of how their framework works on graphs with many more nodes, edges, etc. This reviewer believes, in the current state of this paper, that these experiments do not prove this method's superiority over existing Counterfactual methods nor do it showcase the full behavior of their framework on graphs it is likely to encounter in the real world.

**Essential References Not Discussed:**

n/a

**Experimental Designs Or Analyses:**

For the graph embedding algorithm why did the authors just use a standard backbone of a GCN. It would be interesting to see various architectures trained and how they affect the framework.

The design of the current experiments are correct

**Methods And Evaluation Criteria:**

As stated before, several issues with their experimentation or lack thereof are:
O1. Their is a glaring lack of experiments. It is understandable the authors introduce a novel problem and emphasize that in their experiments however the main metrics are coverage and cost. These experiments are good at showing their methods desired properties but they do not showcase mainstream metrics and experiments such as validity, sparsity, and potentially some other metrics. All these metrics are not necessary but they bolster their framework's strengths.

O2. Another issue is the lack of experiments. There are numerous types of datasets that have not been experimented on. Focusing primarily on molecular data leaves a glaring question for the readers: experimentally how does this framework work on other types of graphs (such as networks, etc.)? Not addressing this question also raises questions of how their framework works on graphs with many more nodes, edges, etc. This reviewer believes, in the current state of this paper, that these experiments do not prove this method's superiority over existing Counterfactual methods nor do it showcase the full behavior of their framework on graphs it is likely to encounter in the real world.

**Other Comments Or Suggestions:**

Please address the weaknesses.

**Other Strengths And Weaknesses:**

Please refer to comments above.

**Questions For Authors:**

Can the authors address why they chose not to include experiments on other types of graphs such as networks, or with a larger scale. Also why did the authors not include more metrics to assess the counterfactual nature of their explanations such as validity. There are several standard metrics in graph interpretability papers that are absent here.

**Relation To Broader Scientific Literature:**

This paper works on generating global counterfactual explanations. Specifically extending this to the notion of common recourses which add several benefits.  Most existing works use local counterfactual explanations so the authors explore a relatively under-explored area.

**Theoretical Claims:**

no issues.

---

> ### Author Rebuttal · Authors · 2025-04-01
>
> >Q1: validity and sparsity metrics are missing.
>
> **Answer:** These two metrics have been considered, but under a different form in the definition of common recourse (in page 2 of our paper).
>
> **Validity:** It measures how often the recourse suggestion changes the model’s prediction. In our definition of recourse we access the oracle, so every counterfactual candidate is a counterfactual, and the validity of all our recourse is $1$.
>
>
> **Sparsity:** It measures how many edges or nodes need to be altered to reach a common recourse.
>
> - We used a normalized GED  in the paper. The sparsity is exactly equal to $Sparsity(G,G') = 1 - Normed GED (G,G')$ for $G,G'$ two graphs. Therefore sparsity is directly embedded in the definition of common recourse, where we explicitly limit the counterfactual to have sparsity at least $1-\theta$, which in our experiments is typically a lower $0.90$ to $0.85$.
>
> -  We use the cost metric in our experiments, defined in Section 2.1, which accounts for the total distance from the covered input graphs to their closest attained counterfactual. Therefore the cost measure an aggregation over $1-Sparsity$.
>
>
> >Q2: Add more datasets, networks, graphs with many more nodes.
>
> **Answer:** As suggested, we provide additional experiments.
>
> We study the IMDB-BINARY and IMDB-MULTI datasets (Yanardag & Vishwanathan, 2015). The IMDB-BINARY features movies from two genres (Action and Romance). The IMDB-MULTI includes movies from three genres (Comedy, Romance, and Sci-Fi). Since our current method is only interested in binary classification, we change the consider the following labels: Comedy and non-Comedy (i.e Romance and Scifi). The parameters for COMRECGC are the same as for the experiments in Section 4 and Table 3.
> *The results show that our method outperforms the baseline GCFExplainer in coverage on both datasets, while preserving a similar cost (higher coverage and lower cost are better).*
>
> **Table 1: Results on the FCR problem for the task of explaining the GIN trained model.**
>
> |  | **IMDB-BINARY** | **IMDB-BINARY** | **IMDB-MULTI** | **IMDB-MULTI** |
> |-|-|-|-|-|
> |  | Coverage | Cost | Coverage | Cost |
> | **GCFExplainer** | 76.5% | 8.33 | 19.9% | **7.65** |
> | **COMRECGC(Ours)** | **80.9%** | **8.10** | **21.9%** | 7.70 |
>
>
>
>
> >Q3: Testing the approach on different GNN architecture
>
>
> **Answer:** As suggested, we provide experiments with different architectures.
>
> We train GAT(Velickovic et al., 2018), GraphSAGE(Hamilton et al., 2017), GIN (Xu et al., 2019) GNN model for a binary classification task, consisting of three convolutional layers, a max pooling layer, and a fully connected layer, following the literature (Vu & Thai, 2020). The model is trained with the Adam optimizer (Kingma & Ba, 2014) and a learning rate of 0.001 for 1000 epochs. The training/validation/testing split is 80%/10%/10%.
>
> The training/validation/testing accuracy tables, alongside the result tables for the experiments are available at: https://anonymous.4open.science/r/COMRECGC-3E4E/tables_additional_experiments.pdf.
>
> We test COMRECGC against the GCFE baseline. The parameters for the experiments and methods are the same as in Section 4 and Table 3 of the paper, in particular $\theta = 0.1$  ($0.15$ for Proteins) and $\Delta = 0.02$.
> **The results show that our method outperforms GCFExplainer in terms of coverage on all datasets for the GIN model explanation, while often offering a lower cost (higher coverage and lower cost are better).**
>
> **Table 2: Results on the FCR problem for the task of explaining the GAT trained model.**
>
> ||**NCI1** | **NCI1** | **Mutagenicity** | **Mutagenicity** |   **AIDS** | **AIDS** | **Proteins** |**Proteins**|
> |-|-|-|-|-|-|-|-|-|
> || Coverage | Cost | Coverage | Cost | Coverage | Cost | Coverage | Cost |
> | **GCFExplainer** | 24.4% | 5.26 | 47.3% | **5.82** | 27.6% | 7.12 | 42.6% | 10.54 |
> | **COMRECGC(Ours)** | **35.6%** | **5.02** | **55.7%** | 6.05 | **30.7%** | **6.89** | **42.9%** | **10.27** |
>
>
> ---
>
> **Table 3: Results on the FCR problem for the task of explaining the GraphSAGE trained model.**
>
> || **NCI1** | **NCI1**  | **Mutagenicity** | **Mutagenicity** | **AIDS** |**AIDS**  | **Proteins** | **Proteins**|
> |-|-|-|-|-|-|-|-|-|
> || Coverage | Cost | Coverage | Cost | Coverage | Cost | Coverage | Cost |
> | **GCFExplainer** | 32.8% | 4.86 | 46.5% | **5.46** | 20.3% | 7.38 | 68.6% | 11.53 |
> | **COMRECGC(Ours)** | **47.9%** | **4.76** | **50.9%** | 5.90 | **21.5%** | **7.16** | **69.4%** | **11.51** |
>
> ---
>
> **Table 4: Results on the FCR problem for the task of explaining the GIN trained model.**
>
> |  | **NCI1** | **NCI1** | **Mutagenicity** | **Mutagenicity** | **AIDS** | **AIDS** | **Proteins** ||
> |-|-|-|-|-|-|-|-|-|
> ||  Coverage | Cost | Coverage | Cost | Coverage | Cost | Coverage | Cost |
> | **GCFExplainer** | 31.2% | 5.13 | 30.4% | **6.05** | 14.7% | 7.68 | 47.3% | 12.21 |
> | **COMRECGC(Ours)** | **45.6%** | **4.58** | **33.7%** | 6.41 | **16.6%** | **7.34** | **48.6%** | **11.32** |

---

> > ### Comment · Reviewer_oYnL · 2025-04-03
> >
> > The authors have addressed most of my concerns, and I will increase the final rating.

---

### Official Review · Reviewer_yQhN · 2025-03-13

**Overall Recommendation:** 4

**Summary:**

In this study, the authors have formalized the problem of generating global counterfactual explanations for Graph Neural Networks (GNNs) with common recourse. Considering the NP-hard nature of the FCR and FC problems, the authors have developed COMRECGC, a method specifically designed to extract high-quality common recourse explanations. Experiments on real-world datasets show that COMRECGC consistently generates global common recourse explanations of significantly higher quality compared to the baseline methods.

**Claims And Evidence:**

The claims made in the submission are supported by clear and convincing evidence.

**Essential References Not Discussed:**

No

**Experimental Designs Or Analyses:**

Yes

**Methods And Evaluation Criteria:**

Yes

**Other Comments Or Suggestions:**

1. Besides GCN, the authors could attempt more GNN architectures to verify the generalizability of the method.
2. The writing of the authors needs to be improved. Since this paper involves many new concepts, the authors should illustrate them by giving more examples.

**Other Strengths And Weaknesses:**

1. The idea of this paper is very novel. The authors formalize the FCR problem. They provide a generalized version of the FCR problem, named FC, and derive an approximation algorithm for a constrained version of FC.

2. Sufficient experimental results have proven the effectiveness of the method proposed by the authors.

3. The authors have made the source code publicly available.

**Questions For Authors:**

1. The authors have analyzed the Complexity Analysis. Then, what is the actual running efficiency of the model?
2. What does "x" in Table 6 represent?

**Relation To Broader Scientific Literature:**

The key contributions of the paper is important to the broader scientific literature.

**Theoretical Claims:**

Yes

---

> ### Author Rebuttal · Authors · 2025-04-01
>
> Thank you for your strong support and encouragement!
>
> >Q1: Testing the approach on different GNN architecture.
>
> **Answer:** We provide the following **additional experiments**:
>
> - **Experiments on different GNN architectures:** GAT, GraphSAGE and GIN on the datasets of the paper for solving the FCR problem.
>
> -  **Experiments on 2 additional datasets, IMDB-BINARY and IMDB-MULTI:** we have compared our method to the GCFE counterfactual mining baseline for solving the FCR problem.
>
> The results and methodology are available in **our response to Reviewer oYnL**, the tables are also available at: https://anonymous.4open.science/r/COMRECGC-3E4E/tables_additional_experiments.pdf.
>
> >Q2: The writing of the authors needs to be improved. Since this paper involves many new concepts, the authors should illustrate them by giving more examples.
>
> **Answer:** We will add examples in the appendix, in particular for the FCR and FC problems introduced.
>
> >Q3: The authors have analyzed the Complexity Analysis. Then, what is the actual running efficiency of the model.
>
> **Answer:** We provide the time complexity analysis of our method in Section 3.4, and the running time in Appendix D.3. We will emphasize the reference to the Appendix in the complexity analysis section.
>
> >Q4: What does ”x” in Table 6 represent?
>
> **Answer:** The Table illustrates raising $\theta$ from $0.1$ to $0.15$, which dramatically increased the number of recourse entering the clustering stage, making DBScan, the clustering algorithm we use for our experiment, intractable. Thank you, we will add this comment to the caption.

---

> > ### Comment · Reviewer_yQhN · 2025-04-02
> >
> > Thank you for your responses.
> >
> > Can you provide the detailed examples of Q2?

---

> > > ### Author Response · Authors · 2025-04-03
> > >
> > > Thank you! We will include the examples in the revised version.
> > >
> > > ### Example for FC problem
> > >
> > > #### **Setup**
> > > - Given a set of input graphs: $\mathbb{G} = \{G_1, G_2, G_3\}$
> > > - And counterfactual graphs: $\mathbb{H} = \{H_1, H_2, H_3, H_4\}$
> > >
> > > - Each counterfactual $H$ is obtained by applying a recourse $f$, i.e a graph transformation, to an input graph $G$ according to the following table:
> > >
> > > ---
> > >
> > > #### **Finding Counterfactuals to Maximize Common Recourse coverage**
> > > - Suppose the recourse transform the input graph as follows:
> > >
> > > | Input Graphs | Recourse | Counterfactual Graphs |
> > > |---------------|--------------|  ---- |
> > > | $G_1$ | $f_1$ | $H_1$|
> > > | $G_1$ | $f_2$ | $H_2$|
> > > | $G_2$ | $f_3$ | $H_3$|
> > > | $G_3$ | $f_1$ | $H_4$|
> > > | $G_3$ | $f_3$ | $H_1$|
> > >
> > > We will write $f_1(G_1) = H_1$.
> > > We say that an input graph, $G \in \mathbb{G}$, is covered by recourse $f$, if: i) both $H$ and $f$ have been chosen within the budget and ii) $F(G) = H$.
> > >
> > > - Given budgets $R = 2$ (recourse) and $T = 2$ (counterfactuals), the goal of the FC problem is to select $2$ counterfactuals in $\mathbb{H}$ that yield the best coverage on $\mathbb{G}$ by using at most $2$ recourse.
> > >
> > > - Suppose we choose counterfactual graphs $H_1$ and $H_3$, then the best two recourse to pick are : $\mathbb{F}_{\mathbb{H}^*} = \{ f_1, f_3 \}$, which allows us to cover the three input graphs as $f_1(G_1) = H_1$, $f_1(G_3) = H_1$ and $f_3(G_2) = H_3$.
> > >
> > > An intuitive way to see this problem is as a "max 2-budget 2-cover problem", where 2-cover means that to "cover" an element, one has to cover it in the two budgets.
> > >
> > > ---
> > > ### Example for FCR problem
> > >
> > > #### **Setup**
> > > - Suppose we have three reject input graphs:
> > >   $\mathbb{G} = \{G_1, G_2, G_3\}$
> > >
> > > - We are given a budget \( R = 2 \), meaning we can choose at most 2 recourse to transform to transform as many graphs as possible into accep graphs.
> > >
> > > #### **Recourse and Coverage**
> > > - Suppose we have a set of possible recourse $\mathbb{F} = \{f_1, f_2, f_3, f_4\}$ where each $f_i$ is an graph modification.
> > >
> > > - Assume the recourse successfuly change the classification for the following subset of the graphs:
> > >
> > > | Recourse Action | Affects Graphs |
> > > |---------------|--------------|
> > > | $f_1$ | $\{G_1, G_2\}$ |
> > > | $f_2$ | $\{G_2, G_3\}$ |
> > > | $f_3$ | $\{G_1, G_3\}$ |
> > > | $f_4$ | $\{G_1, G_2, G_3\}$ |
> > >
> > > - The goal is to choose **$2$** recourse to maximize the number of graphs they apply to. An optimal selection is: $\mathbb{F}^* = \{ f_4 \}$.

---

### Official Review · Reviewer_dyVU · 2025-03-13

**Overall Recommendation:** 3

**Summary:**

The paper introduces COMRECGC, a framework for generating global counterfactual explanations for graph neural networks through common recourse. Unlike local counterfactual explanations which are instance-specific, this approach seeks to find a small set of transformations (recourse) that can convert multiple "reject" graphs into "accept" graphs, thereby providing model-level insights. The authors formalize two novel problem settings - FCR and FC - and prove their NP-hardness. COMRECGC combines a multi-head vertex reinforced random walk to explore the graph edit space for counterfactuals with a clustering approach to identify common recourse patterns. The method is evaluated on four real-world datasets where it substantially outperforms baseline approaches in terms of coverage (the fraction of input graphs explained) while maintaining comparable or lower recourse cost.

## update after rebuttal
I have updated my scores

**Claims And Evidence:**

I found the following main claims in this paper.
1. FCR and FC are Np hard.  these are well supported with theoretical analysis.
2. the proposed method outperforms baseline methods on the FCR and FC problems.  It is supported by experiments. However, only GCN is used.  So it could be more convincing.
3.  They also mention that  the method worth considering for applications such as drug discovery or computational biology.  there is no much evidence like case study to support that.

**Essential References Not Discussed:**

There are some papers not discussed in the paper.

[1] He, Kangjia, et al. "Learning counterfactual explanation of graph neural networks via generative flow network." IEEE Transactions on Artificial Intelligence (2024).
[2] Chhablani, Chirag, et al. "Game-theoretic counterfactual explanation for graph neural networks." Proceedings of the ACM Web Conference 2024. 2024.
[3] Qiu, Dazhuo, et al. "Generating robust counterfactual witnesses for graph neural networks." 2024 IEEE 40th International Conference on Data Engineering (ICDE). IEEE, 2024.
[4] Verma, Samidha, et al. "InduCE: Inductive counterfactual explanations for graph neural networks." Transactions on Machine Learning Research (2024).
[5] Kang, Hyunju, Geonhee Han, and Hogun Park. "Unr-explainer: Counterfactual explanations for unsupervised node representation learning models." The Twelfth International Conference on Learning Representations. 2024.

**Experimental Designs Or Analyses:**

The authors use four benchmark datasets and provide both quantitative and qualitative results.     I have the following concerns.
1. Only a single GNN model (3-layer GCN) is used throughout all experiments. Testing the approach on different GNN architectures (such as GraphSAGE, GAT, or GIN) would have strengthened the generalizability claims of the method.
2. The authors only compare with GCFEXPLAINER and mention that it outperforms other baselines already. However, I think the settings are different, since the results of GCFEXPLAINER on these datasets are different from those in the original paper. So it is not clear whether in the new setting, GCFEXPLAINER is still the best baseline. A direct comparison with other counterfactual explainers under the same experimental conditions would provide stronger evidence.
3. The ablation study in Table 8 effectively demonstrates component contributions but doesn't fully analyze why clustering is more important for some datasets than others. They briefly mention that clustering is "important on large datasets such as MUTAGENICITY, but less so on the smaller PROTEINS." However, the PROTEINS dataset is not significantly smaller than AIDS, yet shows different behavior. This inconsistency is not adequately addressed.
4. The paper uses fixed values for key parameters (θ=0.1 or 0.15, Δ=0.02) across most experiments, with limited sensitivity analysis in the appendix. More extensive parameter tuning would strengthen the robustness claims.

**Methods And Evaluation Criteria:**

Yes. The methods and evaluation approach are suitable for the counterfactual explanation problem.  The multi-head vertex reinforced random walk effectively explores the graph edit space, while the clustering approach sensibly identifies common patterns among recourse.

**Other Comments Or Suggestions:**

I think the factual global methods like XGNN and PGExplainr are also related to this topic.  I suggest the authors also add some discussion on that.

**Other Strengths And Weaknesses:**

Strengths:

1. new prpoblem formulation of the common recourse problem for graph counterfactual explanations.
2. The theoretical framework is well-developed, with clear problem definitions and complexity analysis.
3. effectively combines random walk exploration with clustering to produce useful explanations, and the empirical results demonstrate clear improvements over existing approaches.

Weakness:
1. The organization needs to improve. I think a significant weakness is that the paper is not self-contained. Several important concepts and algorithms are only briefly described with details relegated to appendices, making it difficult to fully understand the method without constantly referring to other sections.  The related work part is also put in the appendix.
2. Some key references are missing
3. experiments are not that comprehensive to support the effectiveness of the proposed method.

**Questions For Authors:**

I don't have specific questions.   Please correct me if my understanding in the above sections is wrong.

**Relation To Broader Scientific Literature:**

The paper builds upon several research threads in the broader explainability and graph neural network literature. Its concept of global counterfactual explanations extends work on local counterfactual explanations for GNNs to the model level, addressing the interpretability gap when dealing with numerous local explanations.
The common recourse framework connects to actionable recourse research but innovates by identifying patterns across instances rather than just instance-specific actions. The multi-head vertex reinforced random walk ia from classic reinforcement random walk.

**Theoretical Claims:**

I briefly checked the proofs. I don't see problems, although some parts are missing, like the analysis of the FCR problem

---

> ### Author Rebuttal · Authors · 2025-04-01
>
> >Q1: Some part of the analysis of the FCR problem is missing.
>
> **Answer:** We will add the following in Appendix B.2:
>
> Let us define $f$ as the function that associates to a set of common recourse its total coverage. We prove that $f$ is submodular:
> Let $A \subseteq B$ be sets of recourse, and let $r$ be a recourse. Suppose $G$ is a counterfactual graph covered by $r$ but not by any recourse in $B$. Since $A \subseteq B$, $G$ is not covered by any elements of $A$. Therefore $f_A(\{r\}) \geq f_B(\{r\})$.
>
>
>
> >Q2: Different GNN architectures.
>
> **Answer:** We provide the following **additional experiments**:
>
> - Experiments on different GNNs: GAT, GraphSAGE and GIN for solving the FCR problem.
>
> -  Experiments on 2 additional datasets, IMDB-BINARY and IMDB-MULTI
>
> The results and methodology are available in **our response to Reviewer oYnL**.
>
>
> >Q3: Comparison with GCFExplainer with incorrect setting.
>
> **Answer:**
> - The results of GCFExplainer are different from the original paper since GCFExplainer aims at identifying global counterfactuals to explain, while the FCR and FC problems consist in finding common recourse, i.e graph transformations, to explain a GNN.
>
> - The comparison between our method and GCFExplainer is valid, because we define counterfactuals and recourse in the same way, with the same values for the parameters $\theta$ and $\Delta$ that define counterfactuals and common recourse.
>
> - We think comparison between our method and typical CE baselines are unfair. A typical CE explainer associates to each reject graph a single CE graph, which is often a subgraph. Both our method and GCFExplainer mine at least 10 times more  counterfactuals, and are not limited to subgraphs. Those counterfactuals are then used to build common recourse through clustering(see Algorithms 1,2,3 page 5 of the paper for more details). Therefore, adding classical CE explainer baselines is not relevant.
>
> >Q4: Table 8.
>
> **Answer:** Thank you! Although both the AIDS and Proteins dataset contain 1837 and 1113 graphs respectively, the number of "reject" graphs is 1473 in AIDS, while it is 366 for Proteins. We are explaining those reject graphs by providing 100 recourse in table 8. So the difference between the two datasets when looking at COMRECGC with or without clustering is possibly explained by the "sparsity" of Proteins reject graphs, and the difficulty to explain the whole dataset in only 100 recourse.
>
>
>
> >Q5: Fixed values for key parameters
>
> **Answer:** You are right! We have two values of each parameters ($\theta=0.1$ or $0.15$, $\Delta=0.02$ or $0.04$).
>
> The parameters $\theta$ and $\Delta$ are crucial in the common recourse definition, and in the FCR and FC problem.
>
> We agree that a finer sensitivity analysis is interesting, specially when the coverage for the NCI1 and Mutagenicity dataset almost double when the common recourse parameter $\Delta$ goes from $0.02$ to $0.04$. The choice of $\theta$ and $\Delta$ is specific to each dataset and application, and we provide a few values to showcase their impact.
>
> A systematic sensitivity analysis is relevant for the overall method and comprehensive experiments will be included in the revision.
>
> We provide new experiments for the sensitivity of $\Delta$. **We observe that a higher $\Delta$ allows for a more comprehensive explanation, as we are allowing similar recourse to be considered "common" more easily. The cost naturally rises as more explanations are covered.**
>
> **Table 1: Results of our method on the FCR problem for the task of explaining the GCN model on the Mutagenicity dataset for different values of $\Delta$.**
>
> | $\theta$       | $\Delta$       | Coverage | Cost  |
> |---------|--------|----------|-------|
> | 0.1     | 0.02   | 51.8%    | 5.63  |
> | 0.1     | 0.025  | 67.15%   | 6.99  |
> | 0.1     | 0.03   | 75.35%   | 7.59  |
> | 0.1     | 0.035  | 83.88%   | 8.03  |
> | 0.1     | 0.04   | 90.0%    | 8.11  |
>
>
>
>
> >Q6: Key references are missing.
>
> **Answer:** We will include the references to the **global factual and local CE literature**:
>
> Recent advancements in counterfactual explanations (CE) for GNNs include He et al.'s (2024) generative flow network, Chhablani et al.'s (2024) game-theoretic approach, and Qiu et al.'s (2024) robust counterfactual witnesses. Verma et al. (2024) introduce InduCE, an inductive method for unseen data, while Kang et al. (2024) focus on unsupervised node representation learning. *While these approaches address local counterfactuals, the task of generating global counterfactual explanations for GNNs remains relatively unexplored.*
>
> >Q7: Paper is not self contained.
>
> **Answer:** We will add the related work section in the main body of the paper. The page limit for the paper is definitely a hard constrain, and we chose to prioritize the definition of the FCR and FC problem, as well as the concept of common recourse in our paper.

---

> > ### Comment · Reviewer_dyVU · 2025-04-02
> >
> > Thank you for the authors' response.
> >
> > I have a follow-up question for Q3.
> >
> > While I understand your point about the differences between typical CE explainers and your approach, I'm still concerned about the limited comparison. Even if traditional CE methods typically provide one counterfactual per graph, couldn't they be adapted for comparison purposes?
> >
> > This type of comparison would provide stronger evidence of your method's advantages over adapted existing approaches. Without such comparisons, it's difficult to fully assess the advancement your method represents beyond GCFExplainer alone.

---

> > > ### Author Response · Authors · 2025-04-02
> > >
> > > Thank you! You are right, we have to perform some modifications. Specifically, One can generate a common recourse explanation from a set of graphs using the clustering algorithm 1 (page 5 in the paper) for **the local counterfactual baselines**. We simply have to filter out the graphs that are not valid counterfactuals. Then the common recourse are formed based on the $\theta$ and $\Delta$ parameters.
> > >
> > > We extend Table 3 in the paper with **additional experiments** with the popular local counterfatual baseline, CF-GNNExplainer \[*Lucic et al., 2022*\]  added:
> > >
> > >
> > > **Table 4: Results on the FCR problem for the task of explaining the GCN trained model for different explainers. The settings are the same as for Table 3 in our paper.**
> > >
> > > | Method  |**NCI1**|**NCI1** | **Mutagenicity** | **Mutagenicity** | **AIDS** | **AIDS**|
> > > |----------------------|----------|------|------------|------|--------|------|
> > > |                      | Coverage | Cost | Coverage   | Cost | Coverage | Cost | Coverage | Cost |
> > > | **CF-GNNExplainer**  | 9.1%     | 7.5  | 12.4%       | 8.90 | 0.1%     | **0.35** |
> > > | **GCFExplainer**     | 21.4%    | 5.75 | 20.6%      | 6.91 | 14.2%    | 6.97  |
> > > | **COMRECGC(Ours)** | **42.9%** | **4.95** | **51.8%**  | **5.63** | **34.7%** | 6.74  |
> > >
> > > **Findings:** The coverage of building common recourse explanation using only CF-GNNExplainer generated counterfactual is noticibly worse than the counterfactual mining methods such as ours. This is likely explained by a number of reasons, such as:
> > >
> > > - Generating less counterfactual graphs (~2k5 graphs for CF-GNNexplainer on Mutagenicity vs 50k+ for GCFExplainer and our method),
> > > - Only considering subgraphs of input graphs and,
> > > - Not taking into account the $\theta$ and $\Delta$ parameters of the CR explanation.
> > > - Please note that GCFExplainer (our main baseline) has been shown to outperform the local counterfactual baselines. And, our method outperforms GCFExplainer.

---

### Official Review · Reviewer_pBUz · 2025-03-14

**Overall Recommendation:** 3

**Summary:**

This paper designs an algorithm COMRECGC for global graph counterfactual explanation (CE). It considers the finding common recourse (FCR) explanation to address the limitations of existing graph CE methods (relying on experts for recourse directions; separating the process of funding recourse direction from data fitting).

The paper includes a complexity analysis, and the code is publicly available. The experiment compares common recourse explanations and graph CE, and shows COMRECGC provide a comparable performance in global CE. Evaluations on benchmark datasets demonstrate strong performance.

## update after rebuttal
I appreciate the authors for their response, which addresses some of my concerns. I stick to the original score of weak accept.

**Claims And Evidence:**

The claims in the submission are generally well supported.

**Essential References Not Discussed:**

N/A

**Experimental Designs Or Analyses:**

The experimental design and analysis appear generally sound, with appropriate methodologies used to validate the proposed approach. However, the improvement in terms of cost does not seem significant.

A more in-depth evaluation of efficiency, such as a detailed comparison of efficiency and robustness with baselines, would strengthen the analysis and further enhance the impact of the results.

**Methods And Evaluation Criteria:**

It would be beneficial for the experiment to include additional evaluations on efficiency, particularly in terms of time and space complexity. Providing a more detailed analysis of computational cost would help assess the practical feasibility and scalability of the proposed approach.

**Other Comments Or Suggestions:**

Few typos are found and perhaps need a further proofread.

**Other Strengths And Weaknesses:**

The point of involving FCR and FC problems is novel, and the source code link is provided.
But in general, the improvement compared with existing GCE baselines does not seem very significant.

**Questions For Authors:**

Could the authors provide any (empirical) evaluation for the model efficiency?

**Relation To Broader Scientific Literature:**

The paper is related to some graph-related scientific discovery and explanation fields.

**Theoretical Claims:**

The paper provides theoretical analysis for the FC problem and the complexity of the proposed method.

---

> ### Author Rebuttal · Authors · 2025-04-01
>
> Thank you for your remarks and we appreciate the support for our work.
>
> >Q1: Could you include additional evaluations on efficiency, particularly in terms of time and space complexity?
>
> **Answer:** We make the following notes:
> * **Time complexity** We have provided the time complexity analysis of our method in Section 3.4 of the paper, and the running time in Appendix D.3.
>
> * **Space complexity:** The space complexity is upper bounded by the number of recourse we are generating through the random walk. The resulting complexity is $O(n|G|)$ where $|G|$ is the number of input graphs and $n$ is the number of top-visited counterfactuals during the random walk. We will add this discussion in the paper.
>
> >Q2: Could the authors provide any (empirical) evaluation for the model efficiency?
>
> **Answer:**
>
> We would like to mention that we have provided time complexity in the paper (see the above answer). If you mean efficiency as effectiveness, we have performed extensive experiments (please Sec. 4 and Appendix D). To strengthen the claim of our model efficiency (or effectiveness) further, we provide the following **additional experiments**:
>
> - **Experiments on different GNN architectures:** GAT, GraphSAGE and GIN on the datasets of the paper for solving the FCR problem.
>
> -  **Experiments on 2 additional datasets, IMDB-BINARY and IMDB-MULTI:** we have compared our method to the GCFE counterfactual mining baseline for solving the FCR problem.
>
> The results and methodology are available in **our response to Reviewer oYnL**, the tables are also available at: https://anonymous.4open.science/r/COMRECGC-3E4E/tables_additional_experiments.pdf.
>
> >Remark 1: Few typos are found and perhaps need a further proofread.
>
> **Answer:** We will go over the paper and fix the typos. Thanks!

---

### Decision · Program_Chairs · 2025-05-01

**Decision:**

Accept (poster)

**Comment:**

Post-rebuttal, all reviewers have converged on an acceptance recommendation, with their primary concerns addressed. After reviewing the paper and rebuttal, I agree that the work makes a meaningful contribution to global graph counterfactual explanation. I recommend acceptance. Please ensure that the necessary revisions are incorporated into the final version.